# Manipulating dehydrogenation kinetics through dual-doping Co$_3$N electrode enables highly efficient hydrazine oxidation assisting self-powered H$_2$ production

Yi Liu[1], Jihua Zhang[2], Yapeng Li[1], Qizhu Qian[1], Ziyun Li[1], Yin Zhu[1] & Genqiang Zhang[1✉]

Replacing sluggish oxygen evolution reaction (OER) with hydrazine oxidation reaction (HzOR) to produce hydrogen has been considered as a more energy-efficient strategy than water splitting. However, the relatively high cell voltage in two-electrode system and the required external electric power hinder its scalable applications, especially in mobile devices. Herein, we report a bifunctional P, W co-doped Co$_3$N nanowire array electrode with remarkable catalytic activity towards both HzOR (−55 mV at 10 mA cm$^{-2}$) and hydrogen evolution reaction (HER, −41 mV at 10 mA cm$^{-2}$). Inspiringly, a record low cell voltage of 28 mV is required to achieve 10 mA cm$^{-2}$ in two-electrode system. DFT calculations decipher that the doping optimized H* adsorption/desorption and dehydrogenation kinetics could be the underlying mechanism. Importantly, a self-powered H$_2$ production system by integrating a direct hydrazine fuel cell with a hydrazine splitting electrolyzer can achieve a decent rate of 1.25 mmol h$^{-1}$ at room temperature.

[1] Hefei National Laboratory for Physical Sciences at the Microscale, CAS Key Laboratory of Materials for Energy Conversion, Department of Materials Science and Engineering, University of Science and Technology of China, Hefei, Anhui 230026, China. [2] Guizhou Provincial Key Laboratory of Computational Nano-Material Science, Guizhou Education University, Guiyang 550018, China. ✉email: gqzhangmse@ustc.edu.cn

The rapidly increasing energy consumption and the deteriorating global environmental concerns have compelled the stringent demand on clean and sustainable energy sources. Hydrogen ($H_2$), as the energy carrier with the highest energy density and zero carbon emission, has been placed much expectation as one of the most fascinating candidates to change the current fossil fuel dominated energy structure[1–4]. Thus, it is one of the central task to exploit green and efficient approaches to produce $H_2$, among which electrocatalytic water splitting is deemed as the suitable technique[5,6]. However, the key challenge originates from the intrinsically sluggish kinetics of anodic oxygen evolution reaction (OER, $4OH^- \rightarrow O_2 + 2H_2O + 4e^-$, 1.23 V vs. RHE) involving four-electron-transfer process[7], which not only lead to the energy wastage, but also further increase the cost due to the utilization of noble metal based electrocatalysts, thus highly limits the large-scale application[8]. Hence, it is greatly desired to develop alternative strategy different from fabricating high performance OER electrocatalysts in order to avoid the high energy consumption at the anode.

Recently, it has been demonstrated to possibly overcome this obstacle by replacing anodic OER with thermodynamically more favorable electrocatalytic oxidation of small molecules, including tetrahydroisoquinoline[9], benzyl alcohol[10], urea[11–14] and hydrazine[15–17], which can hugely decease the cell voltage for $H_2$ production. Among these, hydrazine oxidation reaction (HzOR, $N_2H_4 + 4OH^- \rightarrow N_2 + 4H_2O + 4e^-$) possesses unique feature of tremendously lower theoretical potential of $-0.33$ V (vs. RHE) compared to that of OER (1.23 V vs. RHE)[18]. More importantly, the HzOR coupled $H_2$ production (i.e., overall hydrazine splitting, denoted as OHzS) generates $N_2$ as the only byproduct, which is much safer compared to water splitting producing the mixure of $H_2$ and $O_2$[19,20], as well as enabling the utilization of separator-free electrolyzer. Some pioneering works have achieved inspiring progress regarding the HzOR assisted $H_2$ production[16,21–23]. For example, Sun et al. presented that the $Ni_2P$ nanoarrays grown on Ni foam exhibited superior catalytic activity towards HzOR and could output 500 mA cm$^{-2}$ at a cell voltage of 1.0 V in the two-electrode system[16]. Xia and co-workers reported that the tubular $CoSe_2$ nanosheets grown on Ni foam could act as bifunctional HER and HzOR electrocatalysts, which required a cell voltage of 0.164 V to achieve a current density of 10 mA cm$^{-2}$ in the two-electrode system[21]. Non-precious $Co_3Ta$ intermetallic nanoparticles prepared by Xia's group shows an onset potential of $-86$ mV and two times higher specific activity than commercial Pt/C[24]. Despite these progress, there are still several remained challenges in this area. Firstly, it is still unsatisfactory on the relatively high cell voltage in two-electrode system, especially for high current densities, which makes it desired to develop new materials with excellent catalytic activity for both HER and HzOR. Secondly, at the current infant stage on HzOR electrocatalysis, the theoretical understanding on the underlying mechanism for new materials is undoubtedly necessary for the development of materials chemistry. More importantly, external electric power is required in current OHzS based $H_2$ production, which could be a critical hindrance for practical applications in mobile devices and vehicles. Therefore, the proof-of-concept demonstration of integrated system that can use hydrazine as the sole fuel to produce $H_2$ without external power supply is highly meaningful.

Herein, we present an integrated electrode composed of P, W co-doped $Co_3N$ nanowire arrays in situ grown on nickel foam (denoted as PW-$Co_3N$ NWA/NF) as highly efficient bifunctional electrocatalysts for both HzOR and HER. Remarkably, it can achieve current density of 10, 200, and 600 mA cm$^{-2}$ with required working potential of $-55$, 27, and 127 mV (vs. RHE) for HzOR in 1.0 M KOH/0.1 M $N_2H_4$ electrolyte, which is superior compared to state-of-the-art values[17]. Excitingly, the PW-$Co_3N$ NWA/NF also exhibits Pt-like activity for HER with a low overpotential of 41 mV at 10 mA cm$^{-2}$ and a small Tafel slope of 40 mV dec$^{-1}$, as well as excellent durability in 1.0 M KOH electrolyte. The potential of PW-$Co_3N$ NWA/NF for $H_2$ production is further evaluated as both anode and cathode catalyst for overall hydrazine splitting (OHzS), where an ultrasmall operation voltage of 28 mV is needed to achieve current density of 10 mA cm$^{-2}$, and only 277 mV is required to reach 200 mA cm$^{-2}$, indicating the remarkable results compared with previous literatures[15,16]. Density functional theory (DFT) calculations indicate that the P/W doping can not only largely decrease the free-energy changes of the dehydrogenation of adsorbed $NH_2NH_2$ (denoted as *$NH_2NH_2$), but also make the free energy of adsorbed H ($\Delta G_{H*}$) more thermoneutral compared to pristine $Co_3N$. Furthermore, the proof-of-concept self-powered $H_2$ production system is demonstrated by integrating a direct hydrazine fuel cell (DHzFC) with an OHzS device using PW-$Co_3N$ NWA/NF as the bifunctional catalyst and hydrazine as the sole liquid fuel, with a decent $H_2$ evolution rate of 1.25 mmol h$^{-1}$ at room temperature.

## Results

**Synthesis and characterization of PW-$Co_3N$ nanowire arrays.** The P, W co-doped $Co_3N$ nanowire arrays on Ni foam (PW-$Co_3N$ NWA/NF) are synthesized by a facile two-step process, as indicated in Fig. 1a. Firstly, the precursor nanowire arrays are grown on Ni foam through a low temperature hydrothermal method using phosphotungstic acid (denoted as $PW_{12}$) as the single doping resource for both P and W. Then, the final products can be obtained after thermal annealing treatment in an $NH_3$ atmosphere at elevated temperatures (see details in Methods). The morphology and structures are characterized using field-emission scanning electron microscopy (FESEM), transmission electron microscopy (TEM) and X-ray diffraction (XRD). Figure 1b, c present the panoramic view of PW-$Co_3N$ NWA/NF in different magnifications, which show well-aligned nanowires uniformly and densely grown on Ni foam. The cross-sectional view (Fig. 1d) clearly indicates the twisted feature of the nanowires, which is different from the PW-Co-precursor nanowire arrays with smooth surfaces (Supplementary Fig. 1). The typical TEM image (Fig. 1e) shows that the PW-$Co_3N$ nanowire is composed of interconnected nanoparticles, which could be formed during phase transformation upon the annealing. The high-resolution TEM (HRTEM) image (Fig. 1f) provides clear lattice fringes with interplanar distances of 2.04 and 2.16 Å, respectively, corresponding to the (101) and (002) planes of hexagonal $Co_3N$ phase, which can be further confirmed by the XRD pattern (Supplementary Fig. 2). The P/W doping can be evidenced by element mapping result (Fig. 1g) that exhibits homogeneous distribution of P and W elements throughout the whole nanowire besides Co and N. Figure 1h, i provide aberration-corrected HAADF-STEM images of PW-$Co_3N$ NWA/NF at atomic resolution, with the enlarged image showing the successful doping of W. The W atoms can be seen as bright dots in $Co_3N$ lattice due to Z-contrast in HAADF-STEM image, since the atomic number of W is significantly larger than that of Co. However, the brightness of W atoms is not so obvious due to the relatively thick $Co_3N$ substrate, especially when viewing along the zone axis. We further tilted the TEM holder to make the specimen off the zone axis, so that the lattice of $Co_3N$ cannot be clearly seen, and then the brightness of W showed up (Fig. 1j). The corresponding fast Fourier transformation (FFT) pattern (Fig. 1k) indicates the crystalline $Co_3N$ along the [111] zone axis. The stoichiometric ratio of P: W: Co in PW-$Co_3N$ NWA/NF is about 1:3.5:36.9 calculated from EDS spectrum and

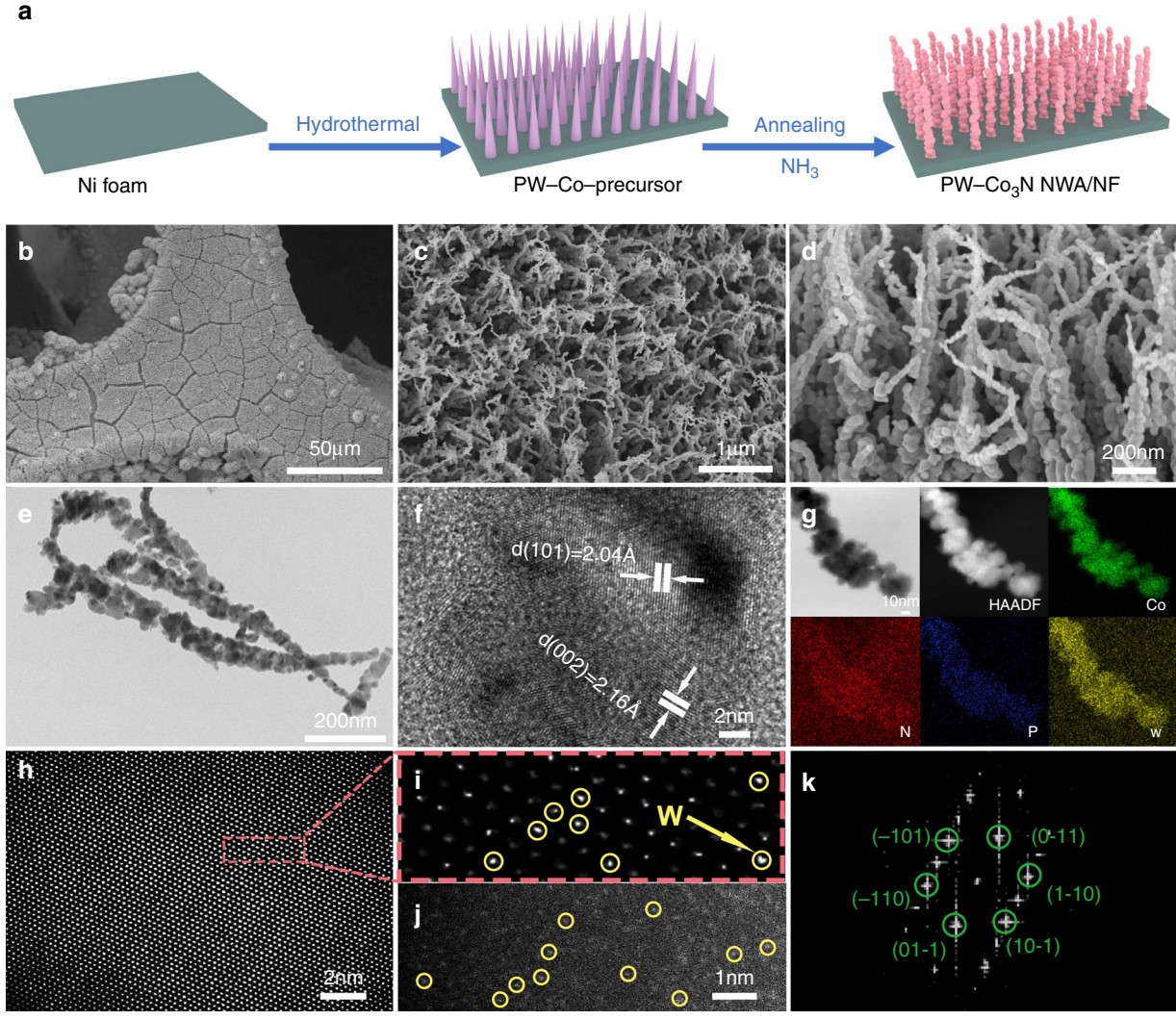

**Fig. 1 Morphological and structural characterization of PW-Co₃N NWA/NF. a** Schematic illustration of the formation process; **b–d** FESEM and **e** TEM images; **f** HRTEM analysis; **g** HAADF-STEM image and corresponding elemental mapping results; **h** aberration corrected HAADF-STEM image; **i** the enlarged picture from (**h**); **j** atomically resolved HAADF-STEM image; **k** the corresponding FFT image of (**h**).

ICP-AES measurement (Supplementary Fig. 3). Moreover, the doping level can be easily tuned by changing the dosage of PW₁₂ during the growth of PW-Co-precursor nanowire arrays, where similar structures can be observed with different doping levels (Supplementary Fig. 4). Moreover, the influence of thermal annealing temperatures on the morphology and crystal structure is systematically investigated. Compared with the sample annealed at 420 °C, lower temperature (350 °C) makes nanowires less orderly while higher temperature (500 °C) gives the similar well-aligned nanowire arrays (Supplementary Fig. 5). However, higher temperature will generate cubic Co₄N phase rather than hexagonal Co₃N formed at 350 and 420 °C (Supplementary Fig. 6)[25].

In order to get insight on the chemical states and further confirm the P/W doping, the comparing investigation of X-ray photoelectron spectroscopy (XPS) characterization on Co₃N NWA/NF and PW-Co₃N NWA/NF are performed. Their XPS survey spectra (Supplementary Fig. 7) notably indicate the existence of P and W signals besides Co and N, further confirming the successful doping. The high-resolution spectrum of Co 2p in PW-Co₃N NWA/NF (Fig. 2a) possesses two peaks at 780.6 and 796.9 eV, respectively, which can be assigned to Co-N bond in Co₃N phase[10,26–29]. Importantly, these two peaks are positively shifted to higher binding energies in PW-Co₃N NWA/

NF. Specifically, the deconvoluted peaks at 778.6 and 793.9 eV can be assigned to metallic Co states in Co₃N[30,31] while the peaks at 786.9 and 803.7 eV can be attributed to satellite peaks[29,32]. Two more peaks located at 782.4 and 799.2 eV could be originated from the surface oxidation, which has also been broadly observed[33,34]. The high-resolution N 1s spectrum (Fig. 2b) can be deconvoluted to metal-N (397.1 eV) and N–H (399.1 eV) peaks, respectively, which further confirm the formation of nitride[35,36]. Interestingly, the binding energy of metal-N displays a negative shift after P/W doping, which is consistent to the positive shift of Co peaks. The W 4f (Fig. 2c) and P 2p (Fig. 2d) spectra can be fitted into W-N (33.3 and 34.8 eV) and P–Co bonds (129.1 eV)[37], respectively, besides the peaks at 35.6, 37.5, and 133.4 eV originated from the surface oxidation[37,38]. These XPS results suggest that the doping of P and W could induce the charge redistribution, and doping could drive the interfacial charge transfer from doped P/W and Co to N, which leads to the peak shift of Co and N[8,10,39]. In order to further confirm this phenomenon, the synchrotron-based X-ray absorption near edge structure (XANES) Co L-edge and N K-edge spectra for PW-Co₃N NWA/NF and Co₃N NWA/N are further collected, as shown in Fig. 2e, f, respectively. It can be observed that two typical L₃ and L₂ peaks of Co L-edge at about 779.3 and 794.9 eV

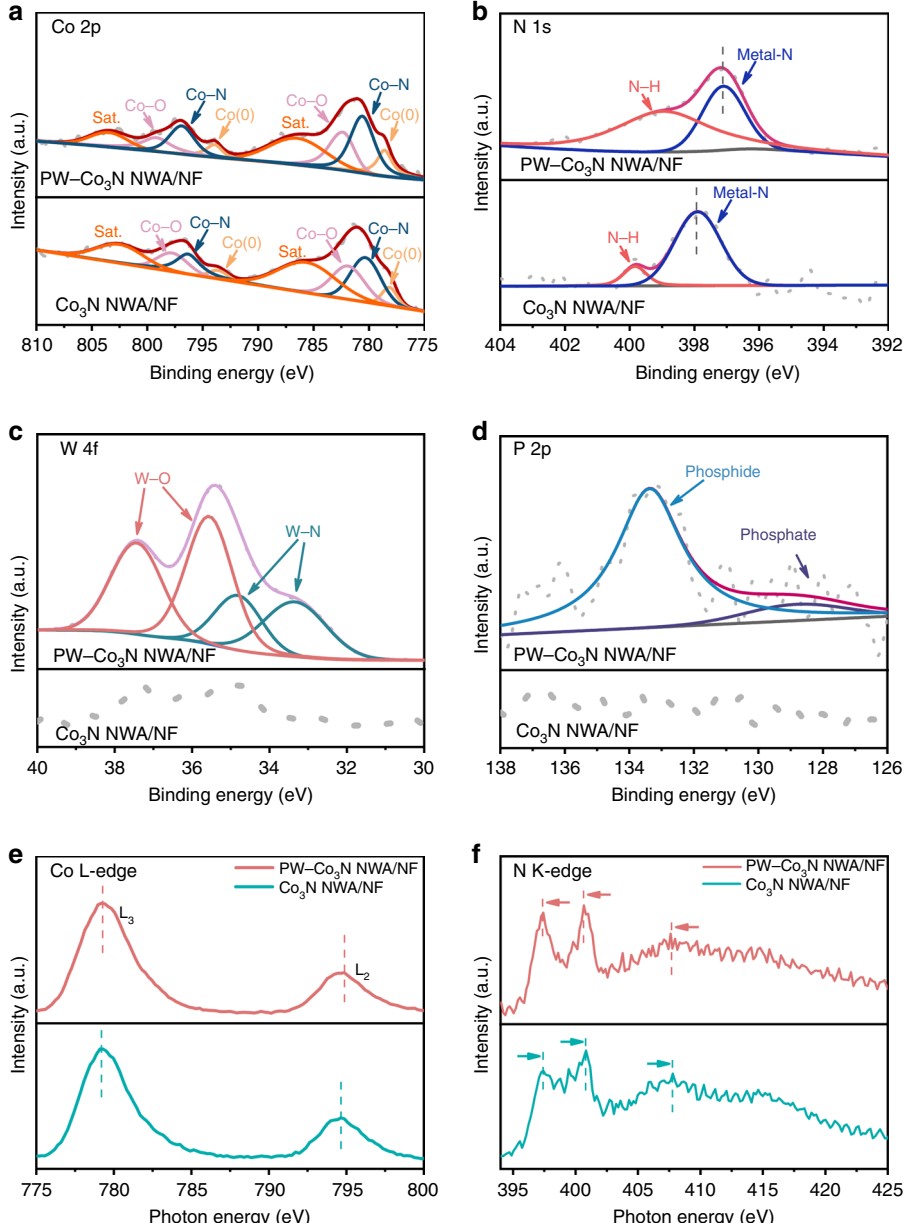

**Fig. 2 High-resolution XPS spectra and soft XANES of PW-Co₃N NWA/NF and Co₃N NWA/NF.** High-resolution XPS spectra of (**a**) Co 2p, (**b**) N 1s, (**c**) W 4f, and (**d**) P 2p; soft XANES of (**e**) Co L-edge spectra; (**f**) N K-edge spectra.

exhibit a positive shift after P/W doping[40], while those of N K-edge spectra show negative shift accordingly[41], which further reveals the possibly existed interfacial charge transfer effect upon doping observed in XPS results.

To disclose the coordination environment of PW-Co₃N NWA/ NF, we further conducted the extended X-ray absorption fine structure (EXAFS) characterization. Figure 3a shows the Co K-edge XANES spectra of PW-Co₃N, Co₃N, Co foil and CoO, where Co foil and CoO are used as the reference. The absorption edge of Co₃N and PW-Co₃N are all located between those of Co and CoO, indicating the average valence of Co should be between Co–Co bond and Co–O bond, which is in accordance with the XPS Co 2p results. In the Fourier transform (FT) of the Co K edge EXAFS spectra (Fig. 3b), there are two typical peaks corresponding to Co–N (~1.3 Å) and Co–Co (~2.3 Å) bonds and the intensity of Co–Co peak decreases obviously in PW-Co₃N compared to pure Co₃N due to the P/W doping. We further performed FT-EXAFS fitting to acquire more information about

the coordination environment of PW-Co₃N and Co₃N and the fitting results are summarized in Supplementary Table 1. Comparing with Co₃N, the Co–Co coordination number in PW-Co₃N increases while the Co-N coordination number decreases, suggesting that the doped W possibly substitutes Co site. Moreover, to more clearly detect the coordination environment in PW-Co₃N and Co₃N, we carried out wavelet transform (WT) of Co K edge EXAFS oscillations due to the high resolution in both K and R spaces. From the WT contour plots of Co₃N (Fig. 3c) and PW-Co₃N (Fig. 3d), the typical Co–Co bond (centered at about 7.5 Å⁻¹) and Co–N bond (centered at about 5 Å⁻¹) are further confirmed. Figure 3e shows the W L₃-edge XANES spectra for PW-Co₃N, W foil and WO₃, in which W foil and WO₃ are used as references. The three samples exhibit peaks at ~10206 eV, where the intensity of PW-Co₃N locates between W foil and WO₃, indicating the successful doping of W. Besides, the FT of W L₃-edge EXAFS spectra (Fig. 3f) declare completely different coordination environment of W in PW-Co₃N compared

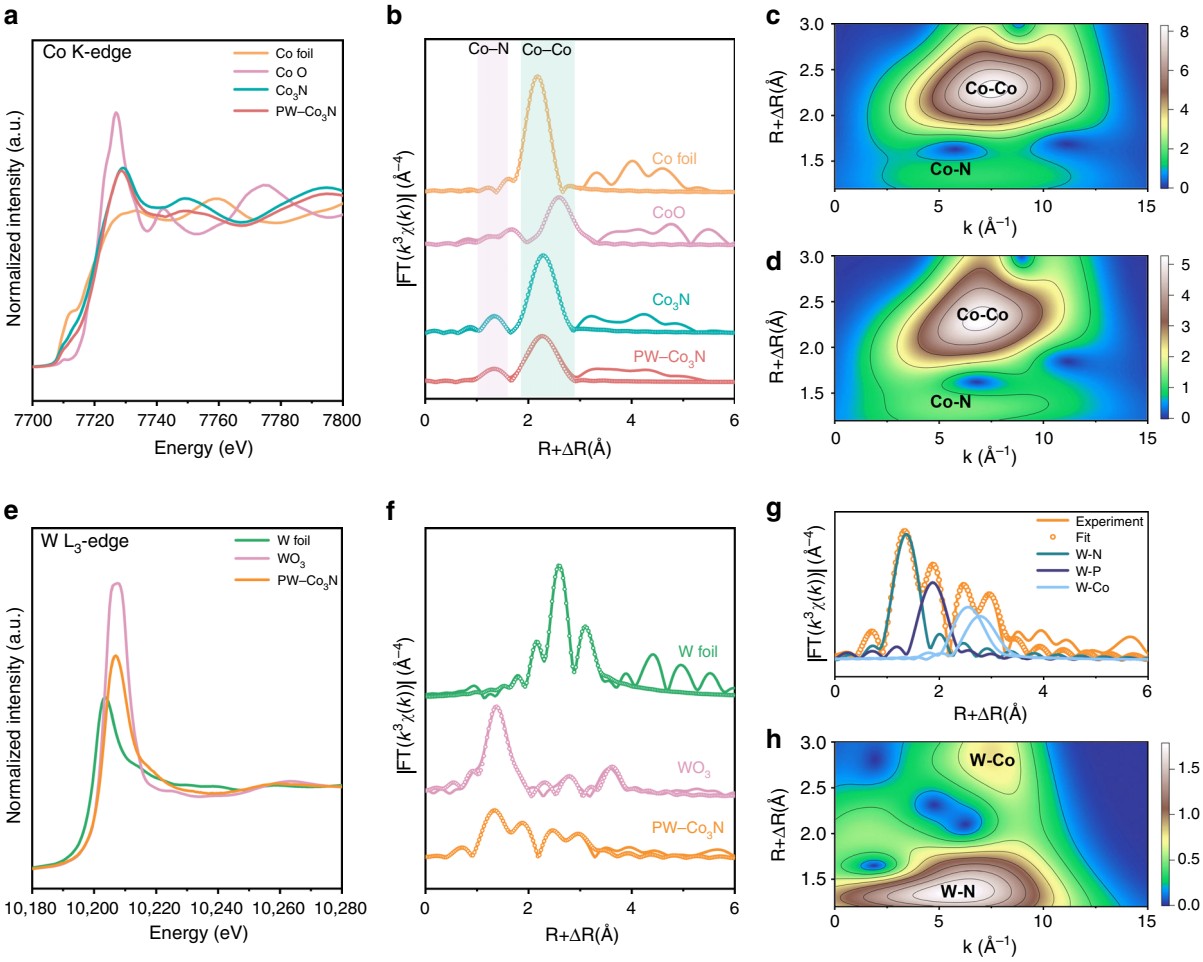

**Fig. 3 XANES and EXAFS spectra of PW-Co₃N NWA/NF and Co₃N NWA/NF. a** The normalized Co K-edge spectra and **b** FT of the Co K-edge EXAFS of PW-Co₃N, Co₃N, Co foil and CoO (line: raw data, scatter: fit); WT of the Co K-edge EXAFS contour plots of (**c**) Co₃N and (**d**) PW-Co₃N; (**e**) The normalized W L₃-edge spectra and (**f**) FT of the W L₃-edge EXAFS of PW-Co₃N, W foil and WO₃ (line: raw data, scatter: fit); (**g**) The FT of the W L₃-edge EXAFS fitting curves of PW-Co₃N; (**h**) WT of the W L₃-edge EXAFS contour plots of PW-Co₃N.

to W foil and WO₃. We also performed FT-EXAFS fitting to probe the coordination environment of doped W in PW-Co₃N, as shown in Fig. 3g, the fitting results are summarized in Supplementary Table 1 as well. The peaks at around 1.35 and 1.87 Å are originated from the W–N bond and W–P bond, respectively, while the peaks at around 2.55 and 2.80 Å could be assigned to W-Co bond. The FT-EXAFS fitting results confirm that the doped W substitutes Co site and bonds with P and N. Moreover, the WT of W L₃-edge contour plots of PW-Co₃N (Fig. 3h) exhibit the typical W-Co bond (centered at about 7.5 Å⁻¹) and Co–N bond (centered at about 6.0 Å⁻¹), which further supports that W substitutes Co site.

**Investigation of electrocatalytic HzOR and HER activities.** The electrocatalytic HzOR activity is evaluated in a typical three-electrode configuration using Hg/HgO (containing 1.0 M KOH solution) as reference electrode and graphite rod as the counter electrode in 1.0 M of KOH/0.1 M of N₂H₄ electrolyte. The optimal doping level and annealing temperatures for PW-Co₃N NWA/NF are firstly examined (Supplementary Figs. 8, 9), where it is confirmed that the sample obtained with the addition of 0.008 mmol PW₁₂ and annealed at 420 °C exhibits the best catalytic activity. We also prepared W-Co₃N NWA/NF and P-Co₃N NWA/NF as control samples (Supplementary Figs. 10–15) to

investigate the role of co-doping. Figure 4a displays the comparing linear sweep voltammetry (LSV) curves of PW-Co₃N NWA/NF, W-Co₃N NWA/NF, P-Co₃N NWA/NF, Co₃N NWA/NF, PW-Co-precursor/NF, bare Ni foam and Pt/C with the N₂H₄ concentration of 0.1 M, which can intuitively indicate the much better electrocatalytic activity of PW-Co₃N NWA/NF compared with others. Specifically, the PW-Co₃N NWA/NF requires ultralow working potentials of −55, 27, and 127 mV to achieve anodic current density of 10, 200, and 600 mA cm⁻², which is far more excellent than that of sole doped W-Co₃N NWA/NF, P-Co₃N NWA/NF and un-doped Co₃N NWA/NF, declaring the indispensability of dual doping. Also, our material PW-Co₃N NWA/NF outperforms most of the reported materials, such as Cu₁Ni₂-N (0.5 mV at 10 mA cm⁻²)[8], Fe⁻CoS₂ (129 mV at 100 mA cm⁻²)[15], Ni₂P/Ni foam (−25 mV at 50 mA cm⁻²)[16] and NiₓP/Ni foam (100 mV at 172 mA cm⁻²)[17], which are also summarized in Supplementary Table 2. The corresponding Tafel plots (Fig. 4b) indicates that the Tafel slope of PW-Co₃N NWA is only 14 mV dec⁻¹, which is much lower than that of W-Co₃N NWA/NF (22 mV dec⁻¹), P-Co₃N NWA/NF (26 mV dec⁻¹), Co₃N NWA/NF (20 mV dec⁻¹), Pt/C (37 mV dec⁻¹), PW-Co-precursor/NF (120 mV dec⁻¹) and bare Ni foam (77 mV dec⁻¹), suggesting the most favorable catalytic kinetics towards HzOR. In order to check the intrinsic nature of the catalytic activity for PW-Co₃N NWA/NF towards HzOR, the LSV curves with different

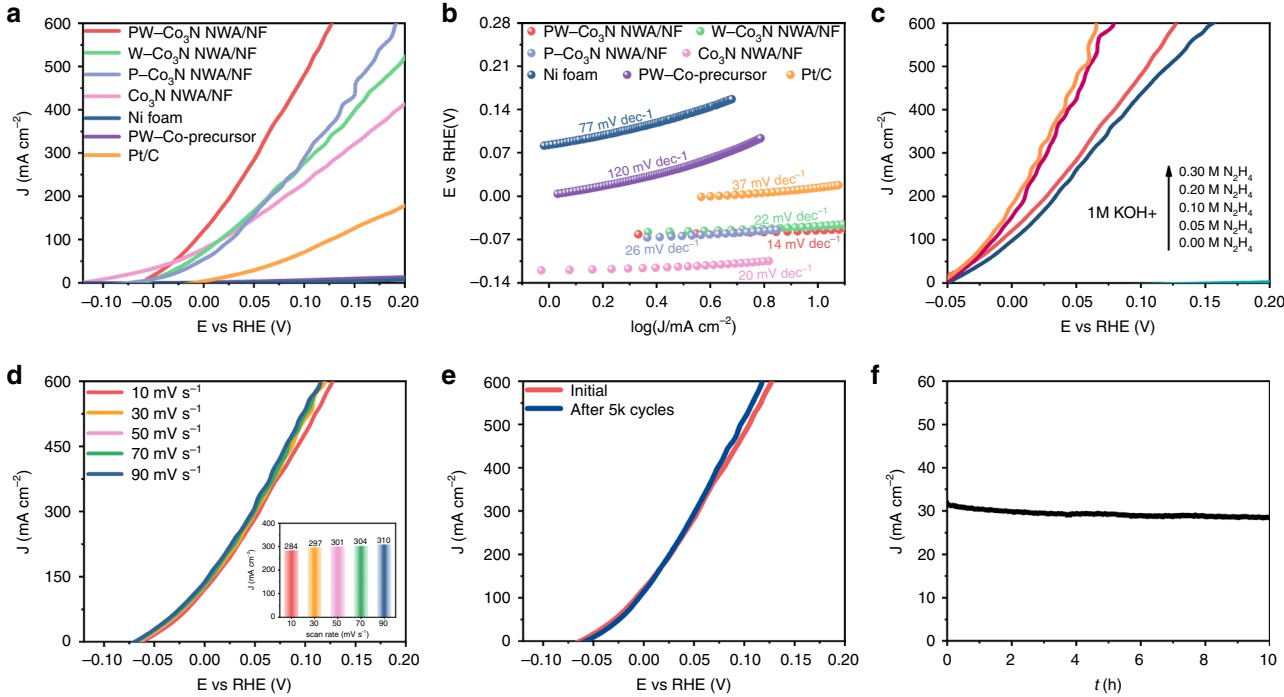

**Fig. 4 Electrocatalytic activity towards HzOR in 1.0 M KOH/0.1 M $N_2H_4$ electrolyte. a** Polarization curves and **b** corresponding Tafel plots of PW-$Co_3N$ NWA/NF, W-$Co_3N$ NWA/NF, P-$Co_3N$ NWA/NF, $Co_3N$ NWA/NF, bare Ni foam, PW-Co-precursor and Pt/C; **c** LSV curves of PW-$Co_3N$ NWA/NF with different concentrations of hydrazine, **d** LSV curves of PW-$Co_3N$ NWA/NF electrodes at different scan rates, and the inset is the corresponding current density at 50 mV vs. RHE for different scan rates, **e** Polarization curves of PW-$Co_3N$ NWA/NF electrode after successive CV test; **f** The chronoamperometric test recorded at working potential of −40 mV.

concentrations of hydrazine are measured, as shown in Fig. 4c. It can be clearly seen that no anodic current can be observed within the tested potential window of −0.05~0.2 V (vs. RHE) without hydrazine. In contrast, the anodic current density sharply rises with the addition of 0.05 M of hydrazine and keeps increasing with increased concentrations. Figure 4d displays the LSV curves of PW-$Co_3N$ NWA/NF with different scan rates ranging from 10 to 90 mV s$^{-1}$, where only slight changes can be observed, suggesting the efficient charge and mass transport process during the catalytic oxidation of hydrazine. This could be consistent with the electrochemical impedance spectroscopy (EIS) results, where only a small charge transfer resistance ($R_{ct}$) of 1.73 Ω can be indicated, much lower than that of $Co_3N$ NWA/NF (2.88 Ω), W-$Co_3N$ NWA/NF (4.96 Ω), P-$Co_3N$ NWA/NF (13.18 Ω), PW-Co-precursor/NF (126.1 Ω), bare Ni foam (723.9 Ω), and Pt/C (25.36 Ω) as shown in Supplementary Fig. 16. As one of the critical factor for practical applications, the durability of the PW-$Co_3N$ NWA/ NF is then evaluated. Figure 4e exhibits the LSV curves after successive cyclic voltammetry (CV) test, which still shows negligible decay after 5000 cycles, indicating excellent stability. Moreover, the long-term stability is further evaluated by the chronoamperometric test at a current density of 30 mA cm$^{-2}$ for 10 h, where 92.3% current retention can be observed (Fig. 4f).

The HER activity of PW-$Co_3N$ NWA/NF electrode is further evaluated in a typical three-electrode cell with 1.0 M KOH as electrolyte. Similarly, the investigation on the optimal conditions are also conducted (Supplementary Fig. 17, 18), and it is found that the sample obtained with identical doping level and annealing temperature to that for HzOR possesses the best catalytic activity for HER, which is denoted as PW-$Co_3N$ NWA/ NF as well unless specified. Then, we compare the LSV curves of the PW-$Co_3N$ NWA/NF and other control samples as shown in Fig. 5a. It can be seen that a small overpotential of 41 mV is required to achieve 10 mA cm$^{-2}$ for PW-$Co_3N$ NWA/NF, which

is only 9 mV larger than that of Pt/C (32 mV) and lower than recent reported transition-metal-nitride-based HER catalysts, such as NiCoN/C nanocages (103 mV)[27], Co-$Ni_3N$ (194 mV)[42], $Ni_3N$/C (64 mV)[43] and NiMoN/NF (56 mV)[44] (see details in Supplementary Table 3). The surface area is an important factor to affect the catalytic activity, we then measured the electrochemical double-layer capacitance ($C_{dl}$) of different materials to compare the electrochemical surface areas (Supplementary Fig. 19). The measured $C_{dl}$ values are 70.0, 46.6, 26.2, 31.4, 1.3, 1.0, and 48.3 mF cm$^{-2}$ for PW-$Co_3N$ NWA/NF, W-$Co_3N$ NWA/ NF, P-$Co_3N$ NWA/NF $Co_3N$ NWA/NF, Ni foam, PW-Co-precursor and Pt/C, respectively, implying the maximum numbers of accessible active sites owned by PW-$Co_3N$ NWA/ NF. In order to evaluate the kinetic behavior, we further analyze the corresponding Tafel plots (Fig. 5b), where a much smaller Tafel slope of 40 mV dec$^{-1}$ can be observed compared to that of W-$Co_3N$ NWA/NF (40 mV dec$^{-1}$), P-$Co_3N$ NWA/NF (143 mV dec$^{-1}$), $Co_3N$ NWA/NF (56 mV dec$^{-1}$), PW-Co-precursor (103 mV dec$^{-1}$) and bare Ni foam (118 mV dec$^{-1}$), and the value was even close to Pt/C (31 mV dec$^{-1}$). To better understand the interfacial charge transfer kinetics, the EIS measurement was conducted to obtain the Nyquist plots (Supplementary Fig. 20), where the PW-$Co_3N$ NWA/NF electrode exhibits the smallest charge transfer resistance ($R_{ct}$) of 2.2 Ω compared to that of control samples, revealing the most excellent interfacial electron transfer kinetics during HER process. Figure 5c indicates the comparing LSV curves between those after continuous CV test and original one, where the same overpotential of 41 mV at 10 mA cm$^{-2}$ can be observed after 5000 cycles, demonstrating the remarkable stability. Besides, the corresponding charge-transfer resistances ($R_{ct}$) shows ignorable changes (the inset of Fig. 5c), implying the structural robustness of the integrated electrode. Furthermore, the PW-$Co_3N$ NWA/NF also exhibits outstanding long-term durability evaluated by the chronoamperometric test

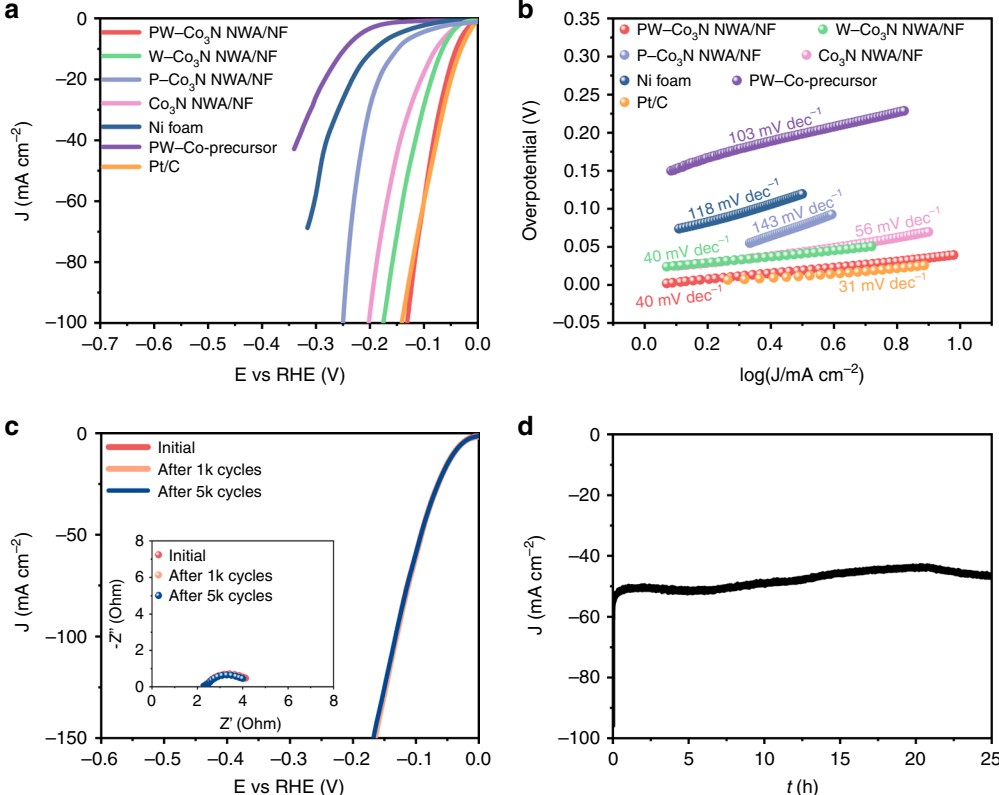

**Fig. 5 Electrocatalytic activity towards HER in 1.0 M KOH electrolyte. a** Polarization curves of PW-Co$_3$N NWA/NF, W-Co$_3$N NWA/NF, P-Co$_3$N NWA/NF, Co$_3$N NWA/NF, Ni foam, PW-Co-precursor and Pt/C towards HER, **b** The corresponding Tafel plots derived from (**a**), **c** Polarization curves of PW-Co$_3$N NWA/NF before and after CV testing of 1000 and 5000 cycles. The inset is corresponding Nyquist plots. **d** The chronoamperometric test recorded at overpotential of 92 mV.

for 25 h at the overpotential of 92 mV (Fig. 5d), where negligible decay can be observed, indicating its high stability for HER. In order to exclude the possible contribution of nickel compounds coming from Ni foam, the control sample by annealing Ni foam with sodium hypophosphite at 420 °C under NH$_3$ (denoted as PN/NF) was synthesized (Supplementary Fig. 21) and its electrocatalytic performance is investigated (Supplementary Fig. 22), where negligible HzOR and HER activity can be detected, verifying the intrinsic excellent HzOR and HER activity of PW-Co$_3$N NWA/NF.

**Evaluation of PW-Co$_3$N NWA/NF as bifunctional catalyst for OHzS.** Inspired by the excellent HzOR and HER catalytic performance, we further demonstrate its potential applications for electrocatalytic H$_2$ production based on OHzS in a two-electrode electrolyzer using 1.0 M KOH added with 0.1 M hydrazine as the electrolyte. Figure 6a shows the comparing LSV curves of OHzS and overall water splitting (OWS) in 1.0 M KOH, in which significantly enhanced energy efficiency can be intuitively seen using hydrazine oxidation assisted H$_2$ production. Specifically, it only requires the overpotentials (compared to the theoretical value of −330 mV) of 358, 428, 501, and 607 mV in OHzS system to reach current densities of 10, 50, 100, and 200 mA cm$^{-2}$ V (Fig. 6b), respectively, while much higher overpotentials (compared to the theoretical value of 1230 mV) of 350, 530, 650, and 869 mV are required in the case of OWS to obtain the same current density. The results proved that our OHzS system not only needs less electric energy in practical use (see more information in Supplementary Table 4), but also exhibits favorable kinetics. Importantly, a high Faradaic efficiency of 96% (Fig. 6c) for H$_2$ evolution in OHzS can be achieved, which is even slightly better

compared to the OWS system (94%, Fig. 6d), exhibiting the excellent efficiency in the two-electrode system. Besides, this two-electrode OHzS system can maintain the current density of 50 mA cm$^{-2}$ with a low cell voltage of 98 mV with acceptable stability considering the hydrazine consumption during 20 h continuous test (Fig. 6e).

**Demonstration of self-powered H$_2$ production.** In order to further demonstrate the possible applications of our catalyst, we built a self-powered system by integrating a direct hydrazine fuel cell (DHzFC) using PW-Co$_3$N NWA/NF anode and commercial Pt/C cathode to drive OHzS for H$_2$ production, which can be schematically illustrated in Fig. 7a. The room temperature performance of the DHzFC shows an open-circuit voltage (OCV) of 0.98 V (Supplementary Fig. 23) and can reach a maximum power density of 46.3 mW cm$^{-2}$ at a cell voltage of 0.429 V (Fig. 7b), which is comparable with the recently reported values under similar working conditions[15,45,46], which is also summarized in Supplementary Table 5. As a proof-of-concept, the self-powered H$_2$ production system is demonstrated by powering the OHzS electrolyzer using the DHzFC with typical images shown in Fig. 7c, d, where vigorous gas evolution can be observed (Supplementary Movie 1). The H$_2$ yield by this system is measured using gas chromatography, as shown in Fig. 7e. It can be calculated that a decent H$_2$ production rate of 1.25 mmol h$^{-1}$ can be achieved at room temperature, which is competitive among the reported values from self-powered H$_2$ production systems[15,47–49], demonstrating the exciting potential for highly efficient H$_2$ productions. Moreover, the total efficiency (TE) from hydrazine to H$_2$ of this self-power system is calculated to be about 45.8%, which is comparable to other hydrogen generation system

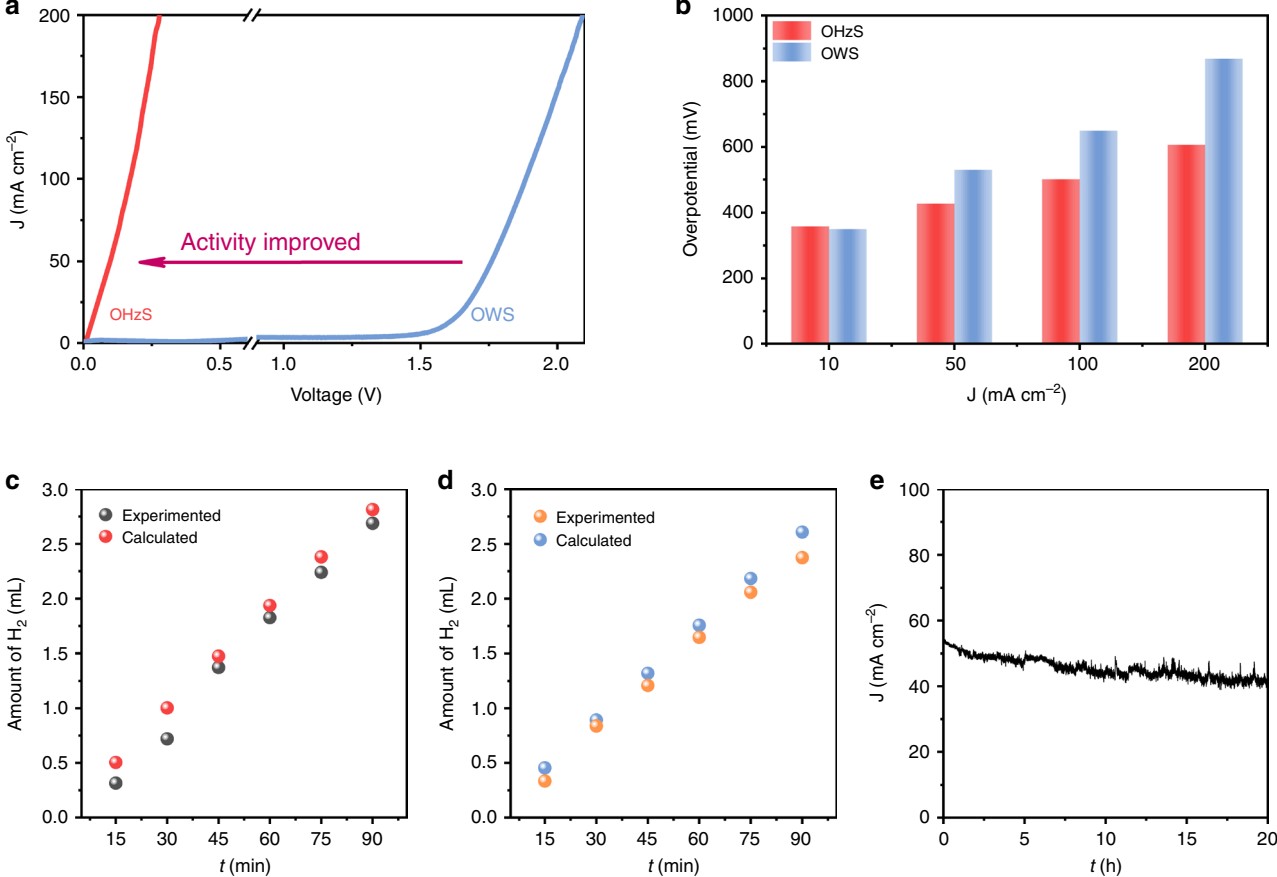

**Fig. 6 Electrochemical performance of OHzS using PW-Co₃N NWA/NF without DHzFC. a** Comparison of LSV curves for OHzS in 1.0 M KOH/0.1 M hydrazine and OWS in 1.0 M KOH using PW-Co₃N NWA/NF as both the anode and cathode, **b** Comparison of the overpotentials required to reach different current densities for OHzS and OWS, **c** The amount of hydrogen theoretically calculated and experimental measured for OHzS, **d** The amount of hydrogen theoretically calculated and experimental measured for OWS, **e** I-t curve of OHzS recorded at 98 mV for 20 h.

(Supplementary Fig. 24 and Supplementary Note 1)[47,50,51]. Compared with gaseous hydrogen or carbon monoxide, hydrazine has the advantage of more convenient transportation and storage as a liquid fuel at ordinary temperatures. However, it is necessary to state that hydrazine is a highly toxic chemical, which may be a challenging issue for the large-scale applications[52]. This is also the major reason that we choose to build our self-powered system working at room temperature and low hydrazine concentration. In order to tackle this obstacle, Asazawa et al.[53] designed a detoxification technique to fix the hydrazine as the carbonyl groups in the harmless, stable and recyclable polymer for storage, which can be released using water or KOH (aq) when needed. This strategy has been considered as one of the promising strategies for large-scale applications involving toxic hydrazine[15,54].

## Discussion
To further recognize the HzOR-active species, we conducted the in situ Raman spectroscopy measurements of PW-Co₃N NWA/NF to obtain real-time detection, as shown in Supplementary Fig. 25. There are two peaks located at 682 and 520 cm⁻¹ for our pristine PW-Co₃N NWA/NF, which is consistent with previous reports[30,55,56]. When the applied potential increased till 0.8 V vs. RHE, the peaks of PW-Co₃N NWA/NF remained constant, which implied that there was no visible surface change during the HzOR process within the measured potential range. The comparing XRD patterns (Supplementary Fig. 25b) of the samples before and after the long-term stability test further confirmed the

unchanged phase. Notably, when the potential increased to 1.7 V vs. RHE, at which the OER process should happen, the original peaks disappeared and the emerging broad peak at the range of 500~600 cm⁻¹ could be assigned to the conversion of metal nitride to metal oxyhydroxide (from Co₃N to CoOOH in our case), which has been broadly observed in previous reports regarding the OER electrocatalysis process[42,44,57,58]. Therefore, it can be concluded that the metal nitride itself could act as the active species for HzOR without observable surface change during the catalytic process.

In order to unravel the possible origin of the better performance after P/W co-doping, the DFT calculations are then applied for both HER and HzOR. Combined with the theoretical calculation and the experimental result regarding the structural stability after doping, it is reasonable to use the model where the surface Co atoms in Co₃N are replaced by W atoms and sub-surface N atoms are replaced by P atoms, as indicated in Fig. 8a–d, Supplementary Fig. 26 and Supplementary Note 2. It has been generally demonstrated that the HER process can be described with three states including an initial pair of H⁺ and e⁻, an intermediate of adsorbed H (H*) and the final product of 1/2 H₂, as shown in Fig. 8e. As the critical descriptor, the free energy of hydrogen absorption ($\Delta G_{H*}$) of PW-Co₃N is calculated to be −0.41 eV, which is closer to the thermoneutral value than that of the pure Co₃N (−0.52 eV). This could imply the facilitated hydrogen adsorption/desorption behavior upon P/W doping[59,60], which is consistent with better HER activity of PW-Co₃N NWA/ NF. In order to understand the fact that the $\Delta G_{H*}$ of PW-Co₃N is

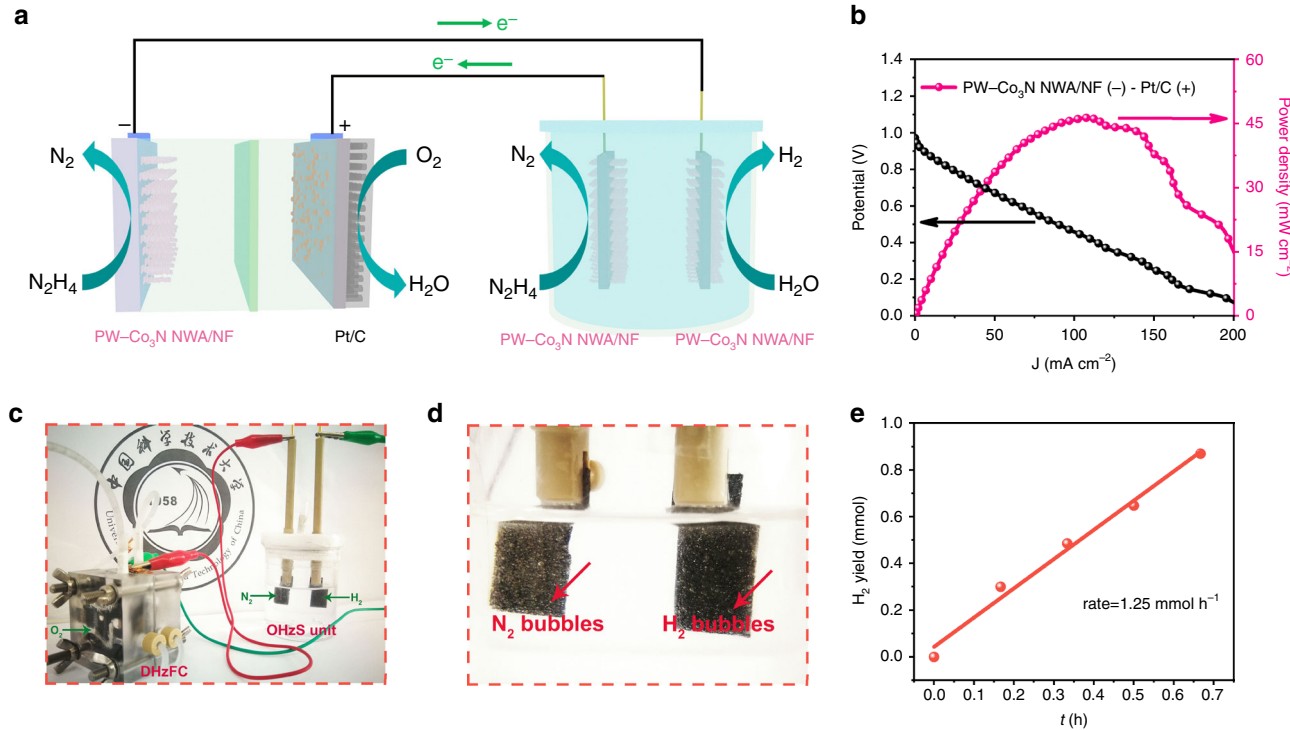

**Fig. 7 Demonstration of self-powered H$_2$ production system. a** Schematic illustration of a self-powered H$_2$ production system integrating a home-made DHzFC and an OHzS unit, **b** Current density (J)-voltage (V) and current density (J)-power density (P) plots for the DHzFC with PW-Co$_3$N NWA/NF anode and Pt/C cathode; **c** Optical image of our home-made self-powered H$_2$ production system, **d** Optical image of gas bubble from the self-powered H$_2$ production system, **e** The H$_2$ generation rate of the self-powered H$_2$ production system in 1.0 M KOH with 0.5 M hydrazine at room temperature.

far more negative compared to benchmark Pt (−0.10 eV) while their HER catalytic activity is almost the same, we further conducted DFT calculations on d-band center analysis, the water absorption energy $\left(\Delta E_{H_2O}\right)$ and charge difference analysis. As shown in Fig. 8f, the d-band center of pure Co$_3$N (−1.41 eV) notably shifts down to (−1.54 eV) after P/W co-doping, which clearly demonstrates that hydrogen desorption is promoted besides the decreasing of hydrogen adsorption energy. In addition, $\Delta E_{H_2O}$ on Co$_3$N (001) and PW-Co$_3$N (001) are −0.31 and −0.56 eV (Fig. 8g), which indicates that the adsorption of H$_2$O molecule is energetically more favorable on the PW-Co$_3$N surface. These results suggest that the P/W doping can not only manipulate the hydrogen absorption energy, but also tailor the d-band center and facilitate the absorption of H$_2$O, which can synergistically promote the HER activity. We further performed the charge density difference plot of PW-Co$_3$N, as shown in Fig. 8h and i. As can be seen, the charge redistribution is dominantly restricted on P/W and their nearest neighboring Co/N, and the doped W and nearby Co sites possess charge depletion while P and the neighboring N possess charge accumulation. This electron localization behavior upon doping could contribute to the enhanced catalytic activity of PW-Co$_3$N.

The theoretical deciphering of HzOR process is more important since its investigation is relatively rare compared to that of HER process. Therefore, DFT calculations on the free energy of N$_2$H$_4$ adsorption ($\Delta G_{N2H4*}$) and each dehydrogenation step from adsorbed NH$_2$NH$_2$* to N$_2$ on the Co site of the (001) surface of Co$_3$N and PW-Co$_3$N with Co-termination are further conducted, as shown in Supplementary Fig. 27 and Fig. 8j. Supplementary Fig. 27 exhibited two different unit cells of Co$_3$N, i.e., Co-terminated and N-terminated. The distance between N$_2$H$_4$ molecular and the nearest neighboring N atom is 5.39 Å, which is much larger than the distance of 1.98 Å between N$_2$H$_4$ and Co

atom. This means that the N atom terminated unit cell is not favorable for N$_2$H$_4$ adsorption, which is inconsistent with our experimental results. Therefore, we adopted the Co-termination unit cell as the model for DFT calculation. As indicated, the PW-Co$_3$N exhibits a more negative $\Delta G_{N2H4*}$ of −1.12 eV compared to that of Co$_3$N (−0.35 eV), suggesting the more favorable N$_2$H$_4$ adsorption, which is undoubtedly important for the further catalytic oxidation process. More importantly, based on the calculation results on the comparison of free-energy changes of each dehydrogenation step for both Co$_3$N and PW-Co$_3$N (Fig. 8j), it can be concluded that the dehydrogenation of *NHNH to *N$_2$H is the potential determining step (PDS) for Co$_3$N towards HzOR, while the PDS for PW-Co$_3$N is the desorption process of *N$_2$ to N$_2$. As indicated, a much higher $\Delta G$ value of +0.7 eV is obtained for pure Co$_3$N, while it is only +0.5 eV for PW-Co$_3$N. These results demonstrate that the introduction of P and W could largely optimizes electronic structure of Co$_3$N, hence facilitates the thermodynamic behavior of both hydrogen adsorption in HER and dehydrogenation process in HzOR.

In summary, we have demonstrated that the integrated electrode composed of P, W co-doped Co$_3$N nanowire arrays grown on Ni foam (denoted as PW-Co$_3$N NWA/NF) possesses benchmark electrocatalytic activity toward both HzOR and HER. Specifically, it can achieve 10, 200, and 600 mA cm$^{-2}$ with ultralow working potential of −55, 27, and 127 mV (vs. RHE) for HzOR in 1.0 M KOH/0.1 M N$_2$H$_4$ electrolyte. The PW-Co$_3$N/NF also exhibits Pt-like activity for HER with a low overpotential of 41 mV at 10 mA cm$^{-2}$ and a small Tafel slope of 40 mV dec$^{-1}$ in 1.0 M KOH electrolyte. Importantly, with the built two-electrode electrolyzer using PW-Co$_3$N NWA/NF as both anode and cathode catalysts for overall hydrazine splitting (OHzS), only an operation voltage of 28 mV is needed to achieve the current density of 10 mA cm$^{-2}$, and only 277 mV is required to reach 200 mA cm$^{-2}$, which is superior compared to state-of-the-art

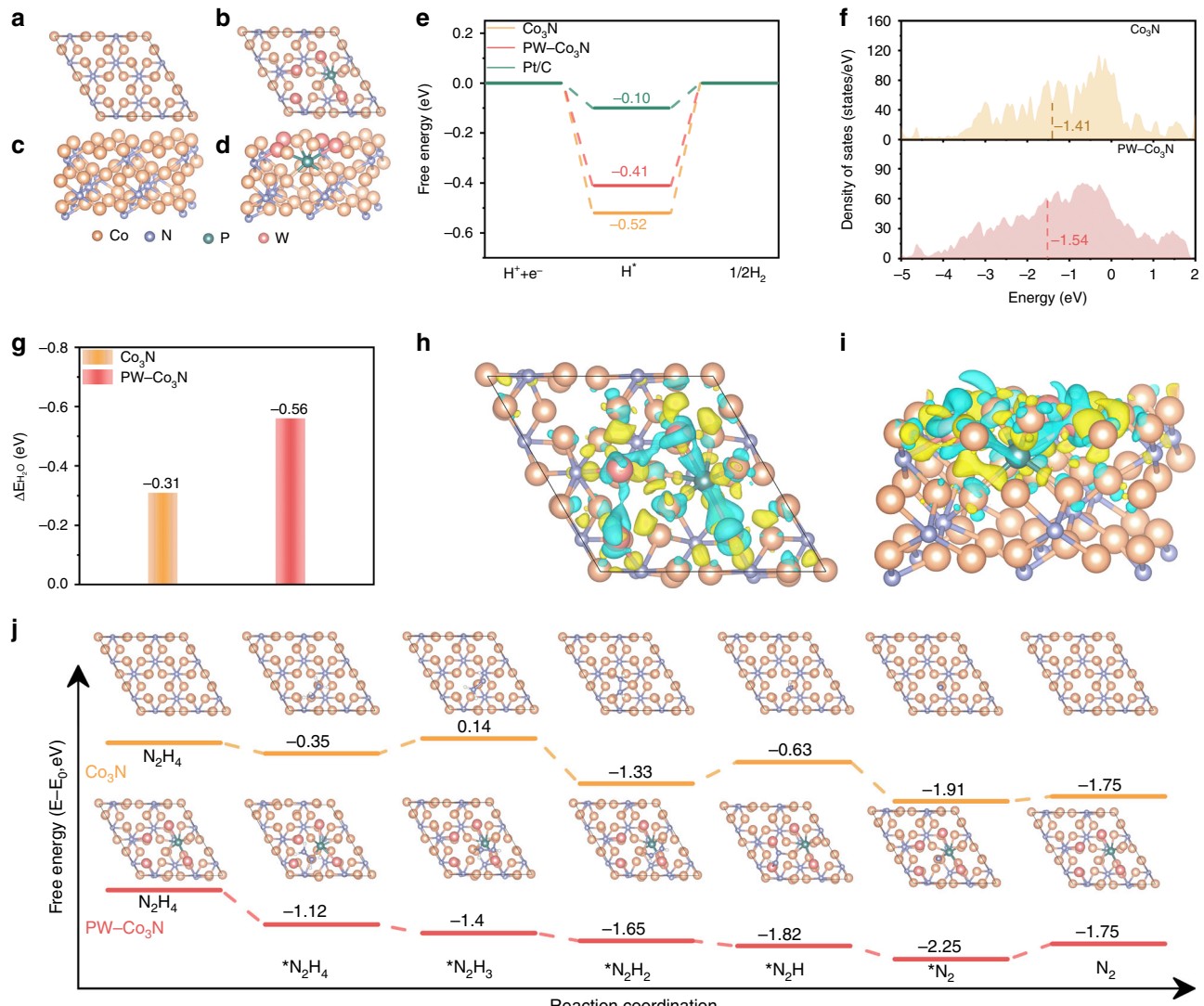

**Fig. 8 DFT calculated profiles of free energy.** Top- and side- view of atomic structure models for (**a**, **c**) $Co_3N$; (**b**, **d**) PW–$Co_3N$; **e** The free-energy diagram of HER at the equilibrium potential for $Co_3N$ and PW–$Co_3N$, H* denotes that intermediate adsorbed hydrogen; **f** The d band of density of states (DOS) of $Co_3N$ and PW–$Co_3N$; **g** Water adsorption energy on $Co_3N$ and PW–$Co_3N$; Top-(**h**) and side-(**i**) view of charge density difference analysis for PW–$Co_3N$ with the cyan region representing charge depletion and the yellow region representing charge accumulation. The isosurface value is 0.012 e$Å^{-3}$; **j** The free-energy profiles of HzOR on the $Co_3N$ and PW–$Co_3N$ surfaces. The inset in (**j**) are the most stable configurations of the each adsorbed intermediate on the Co site.

values. Furthermore, as a proof-of-concept, the self-powered $H_2$ production system is demonstrated by using a homemade direct hydrazine fuel cell (DHzFC) to drive OHzS with hydrazine as the sole liquid fuel. The DFT calculations provide the fundamental origins of the enhanced HzOR and HER performance, in which the P/W doping can not only decrease the free energy of the PDS, i.e., dehydrogenation of NHNH* to NHN*, but also make the $\Delta G_{H*}$ more thermoneutral.

## Methods

**Chemical and materials**. Cobalt nitrate hexahydrate ($Co(NO_3)_2 \cdot 6H_2O$), urea, ammonium fluoride ($NH_4F$), sodium tungstate dihydrate, sodium hypophosphite and phosphotungstic acid hydrate ($H_3PW_{12}O_{40} \cdot xH_2O$, $PW_{12}$) were purchased from Aladdin Industrial Corporation. All of the chemicals were used directly without further purification.

**Synthesis of PW-$Co_3N$ NWA/NF**. The PW-$Co_3N$ nanowire arrays are synthesized via a simple two-step method. In a typical synthesis, 1.9 mmol of Co ($NO_3$)$_2 \cdot 6H_2O$, 10.0 mmol of urea, 4.0 mmol of $NH_4F$ and 0.008 mmol of $PW_{12}$ are dissolved in 40 mL of deionized water, firstly. The as-obtained solution is continuously stirred to achieve the uniform dispersion and then transferred into a

50 mL Teflon-lined autoclave. A piece of Ni foam (3 cm × 2 cm) is pretreated with 3.0 M of HCl to remove the possible oxidation layer and impurity on the surface, and then washed with DI water and ethanol before use. Then, the cleaned nickel foam is subsequently immersed into the autoclave and heated at 120 °C for 10 h. After cooling to room temperature, the as-obtained PW-Co-precursor grown on nickel foam is taken out and washed with DI water and ethanol for several times. The final PW-$Co_3N$ nanowire arrays can be obtained by the thermal nitridation process at 420 °C for 2 h under $NH_3$ atmosphere. The doping level can be easily adjusted by simply changing the added amount of $PW_{12}$ while keeping other conditions identical. The W-$Co_3N$ NWA/NF was prepared under the same synthesis route except replacing $PW_{12}$ by sodium tungstate dihydrate with the same added amount of 0.1 mmol for tungsten. Besides, P-$Co_3N$ NWA/NF was synthesized by the conventional vapor phosphorization method at 300 °C under Ar atmosphere using a spot of sodium hypophosphite as the doping agent (5 mg sodium hypophosphite and a piece of 2 cm* 3 cm as-prepared $Co_3N$ NWA/NF).

**Materials characterization**. The morphology characterization was tested by field-emission scanning electron microscopy (FESEM, JSM-6700F, SU-8200, Sirion200) and transmission electron microscopy (TEM, JOEL, JEM-2010, JEM-ARM200F, HT-7700). The powder X-ray diffractor (XRD, TTR-III, Japan) was used for analyzing the crystal structure of as-obtained samples. Valence state of element of the samples were obtained by X-ray photoelectron spectroscopy (XPS, ESCALAB 250). The soft XANES (Co L-edge and N K-edge spectra) were performed on Photoemission Endstations (BL10B) in National Synchrotron Radiation Laboratory

(NSRL). The XANES and EXAFS of Co K-edge and W $L_3$-edge spectra were collected on the beamline BL01C1 in NSRRC. XANES and EXAFS data reduction and analysis were processed by Athena software. In situ potential-dependent Raman spectroscopy measurement towards HzOR were under programmed applied potentials in 0.1 M KOH/20 mM $N_2H_4$. The electrocatalyst is held for 120 s at the intended potential to reach steady state conditions before recording each spectrum. Raman spectroscopy was conducted by Raman spectrometer (Renishaw inVia) with a 532 nm excitation laser.

**Electrochemical measurements**. Electrochemical performances of the samples were evaluated by using an electrochemical workstation (CHI Instruments 660E, China) at room temperature. In a typical three-electrode system, the as-prepared sample was used as work electrode directly with Graphite rod (6 mm in diameter) as counter electrode and Hg/HgO (1.0 M KOH) as reference electrodes in alkaline aqueous solution, respectively. EIS spectra were recorded with operating over-potentials set at −0.05 V for HzOR and −0.1 V for HER. The frequency ranges from 100 000 to 0.1 Hz. All potentials are given versus reversible hydrogen electrode (RHE) according to the equation: E (vs. RHE) = E (vs. Hg/HgO) + 0.059 × pH + 0.098. All of the data were IR-corrected by the equation: $E_{compensated} = E_{measured} − IR_s$ ($R_s$ is the solution resistance according to EIS test). As for the DHzFC, Nafion 115 membranes were used as the solid electrolyte and the anodes were the PW-$Co_3N$ NWA/NF with Pt/C (20 wt.%) loaded on carbon paper as cathodes. When the DHzFC was testing, an aqueous solution containing 1.0 M KOH and 0.5 M $N_2H_4$ was added into the anode side with a flow rate of 1.2 mL min$^{-1}$ by a silicone tube and a peristaltic pump. Oxygen was plunged into the cathode side as well.

**Computational details**. DFT based first-principles calculations are performed using the projected augmented wave (PAW)[61] method implemented in the Vienna ab initio simulation package (VASP)[62,63]. The Kohn-Sham one-electron states are expanded using the plane wave basis set with a kinetic energy cutoff of 500 eV. The revised-Perdew-Burke-Ernzerhof (RPBE)[64] exchange-correlation functional within the generalized gradient approximation (GGA) is employed. As the active surface, the $Co_3N$ (001) surface is modeled by a periodic six-layer slab repeated in 2 × 2 surface unit cell with Co-termination. Four of the surface Co atoms are substituted by four W atoms to model the W-doped $Co_3N$, while in this W-doped $Co_3N$ model, the P atom substitute one of the surface N atoms to model the W, P-codoped $Co_3N$. The Brillouin-zone (BZ) integration is carried out using the Monkhorst-Pack[65] sampling method with a density of 3 × 3 × 1 for the geometry optimizations. A vacuum layer of 15 Å is included to avoid the interaction between neighboring slabs. All atoms are fully relaxed until the maximum magnitude of the force acting on the atoms is smaller than 0.03 eV/Å.

The oxidation of hydrazine into nitrogen and hydrogen occurs in the following six consecutive elementary steps:

$$(A)^* + N_2H_4 \rightarrow {}^*N_2H_4, \tag{1}$$

$$(B)^*N_2H_4 \rightarrow {}^*N_2H_3 + H^+ + e^-, \tag{2}$$

$$(C)^*N_2H_3 \rightarrow {}^*N_2H_2 + H^+ + e^-, \tag{3}$$

$$(D)^*N_2H_2 \rightarrow {}^*N_2H + H^+ + e^-, \tag{4}$$

$$(E)^*N_2H \rightarrow {}^*N_2 + H^+ + e^-, \tag{5}$$

$$(F)^*N_2 \rightarrow {}^* + N_2. \tag{6}$$

The asterisk (*) represents the reaction surface of these calculated $Co_3N$(001), W-doped $Co_3N$(001), and W, P-codoped $Co_3N$(001). $^*N_2H_4$, $^*N_2H_3$, $^*N_2H_2$, $^*N_2H$, and $^*N_2$ denote the models with the corresponding chemisorbed species residing in the reaction surfaces. Among these six elementary steps, steps (A) and (F) are the adsorption of $N_2H_4$ and desorption of $N_2$, respectively. The other four elementary steps involve the generation of one proton and one electron. Then, using the computational hydrogen electrode (pH = 0, p = 1 atm, T = 298 K)[66], the Gibbs free energy of $H^+ + e^-$ was replaced implicitly with the Gibbs free energy of one-half a $H_2$ molecule. Thus the reaction Gibbs free energies can be calculated with eqs:[67]

$$\Delta G_A = \Delta G_{^*N_2H_4} − \Delta G_* − \Delta G_{N_2H_4} \tag{7}$$

$$\Delta G_B = \Delta G_{^*N_2H_3} + 0.5\Delta G_{H_2} − \Delta G_{^*N_2H_4} − eU − kTln10^*pH \tag{8}$$

$$\Delta G_C = \Delta G_{^*N_2H_2} + 0.5\Delta G_{H_2} − \Delta G_{^*N_2H_3} − eU − kTln10^*pH \tag{9}$$

$$\Delta G_D = \Delta G_{^*N_2H} + 0.5\Delta G_{H_2} − \Delta G_{^*N_2H_2} − eU − kTln10^*pH \tag{10}$$

$$\Delta G_E = \Delta G_{^*N_2} + 0.5\Delta G_{H_2} − \Delta G_{^*N_2H} − eU − kTln10^*pH \tag{11}$$

$$\Delta G_F = \Delta G_* + G_{N_2} − \Delta G_{^*N_2} \tag{12}$$

U and the pH value in this work is set to zero. The adsorption or reaction Gibbs free energy is defined as: $\Delta G = \Delta E + (ZPE − T\Delta S)$, where $\Delta E$ is the adsorption or reaction energy based on DFT calculations, $\Delta ZPE$ is the zero point energy (ZPE)

correction, T is the temperature, and $\Delta S$ is the entropy change. For each system, its ZPE can be calculated by summing vibrational frequencies over all normal modes ν (ZPE = $1/2\Sigma\hbar\nu$). The entropies of gas phase $H_2$, $N_2$, and $NH_2NH_2$ are obtained from the NIST database[68] with standard condition, and the adsorbed species were only taken vibrational entropy (Sv) into account, as shown in the following formula:

$$Sv = \sum_i R\{h\nu_i/[k_B T^*exp(h\nu_i/k_B T) − k_B T] − ln[1 − exp(−h\nu_i/k_B T)]\} \tag{13}$$

Among which $R = 8.314$ J mol$^{-1}$ K$^{-1}$, $T = 298.15$ K, $h = 6.63 \times 10^{-34}$ J s, $k_B = 1.38 \times 10^{-23}$ J K$^{-1}$, i is the frequency number, $\nu_i$ is the vibrational frequency (unit is cm$^{-1}$).

## Data availability
The authors declare that the main data that support the findings of this study are included within the article and Supplementary Information part. Extra data are available from the corresponding author upon reasonable request.

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

## Acknowledgements

G.Q.Z. acknowledges the financial support from the National Natural Science Foundation of China (Grant No. 21601174), the Recruitment Program of Global Experts and the Fundamental Research Funds for the Central Universities (WK2060190081). The numerical calculations in this paper have been done in the Supercomputing Center of University of Science and Technology of China and TianHe-2 at LvLiang Cloud Computing Center of China. J.H.Z. acknowledges the Natural Science Foundation from science and technology department of Guizhou Province (Nos. QHPT[2017]5790-02).

## Author contributions

G.Q.Z. conceived the idea and supervised this project. Y.L. and J.H.Z. contributed equally to this work. Y.L. conducted the project. J.H.Z. performed the DFT calculations. Y.P.L., Q.Z.Q., Z.Y.L., and Y.Z. helped in materials synthesis and electrochemical test. G.Q.Z. and Y.L. co-wrote the manuscript.

## Competing interests

The authors declare no competing interests.
