## [Peer Review File · Nature Communications]

Reviewers' comments:

Reviewer #1 (Remarks to the Author):

This manuscript reports the use of P and W co-doped Co₃N for hydrogen evolution and hydrazine oxidation. The catalyst shows good performance. The analysis of the manuscript needs significant improvement, especially for providing the physical insights why P and W co-doped yields good catalysts. I would recommend publication of the manuscript with major revision. My detailed comments are as follows:

1. Only P and W co-doping are reported. Have the authors tried only P doped or only W doped samples? That would provide deep insights in the role of co-doping.
2. All the results were obtained using related materials (PW co-precursor, Co₃N, Ni foam) to the current co-doped catalyst, it would be good to use some other known catalysts such as Pt to provide an extra external reference.
3. On page 13, can the authors comment on why the electrochemical surface area for PW-Co₃N is so much bigger or have more number of accessible active sites?
4. The authors cited "best reported values" in many places. It would be much more informative if they listed a few concrete values and systems.
5. On page 16, it mentioned "energetically favorable to replace surface Co atoms ...". Please provide those energetic data.
6. While the hydrogen adsorption free energy becomes less negative in the co-doped case, it is still far from the conventional optimal value of 0. Even pure Ni has free energy of ~ -0.3 eV. It is not obvious from fig 7b why the catalyst has favorable HER activity
7. N in figure 7f is hard to see. Please consider revise the color scheme.
8. I would recommend performing some charge analysis, in the form of calculating Bader charges or obtaining charge density difference plot, to understand what is the impact of co-doping on Co₃N.
9. Significant justification is needed for the co-doping calculation model. Why the Co-termination unit cell is adopted? What is the current W coverage? Why using 4 W but only one P? Why only surface substitution is used, but not subsurface ones?

Reviewer #2 (Remarks to the Author):

This manuscript by Liu and co-workers described P, W co-doped cobalt nitride nanowire arrays for catalytic hydrazine oxidation and hydrogen evolution. In general, the resulted nanocomposites demonstrated quite impressive activities for both reactions, which allowed the integration of a direct hydrazine fuel cell with a hydrazine splitting electrolyzer for self-powered H₂ production. However, the manuscript contains substantial unproven or unnecessary claims and its overall quality does not warrant its suitability for publication in leading journals such as Nature Commun.

1. The fabrication method is quite routine, and there are many hundreds of similar works to decorate various electrochemically active materials onto Ni foams for various electrocatalytic reasons such as OER, HER, CO₂RR, N₂RR, and HzOR. And this work is no exception.
2. The structure characterization is very poor. There is no discussion of how and where P and W species locate within the Co₃N lattice. HRTEM imaging and X-ray absorption seem necessary to address this critical issue. The authors did not even provide the chemical composition of their materials system. Without all these key information, what is the reliability for their DFT models?
3. The reviewer did not see the particular advantage of so called self-powered H₂ production system, because direct splitting of hydrazine is very straightforward to produce H₂ and N₂ within the scope of heterogeneous catalysis. Besides, this self-powered H₂ production system has been reported by the authors of Ref. 15. So essentially there is nothing new here.

4. By claiming "a record high", "outperforms most of the literatures", "is one of the best reported values", "is superior compared to the reported values", "the exciting potential for highly efficient H₂ productions", "possesses benchmark electrocatalytic activity", "is superior compared to state-of-the-art values", the authors should at least provides sufficient literatures for direct comparison (such as a table?). Unfortunately, the authors did not even conduct control experiments using standard electrode materials such as Pt/C, which makes the data less convincing. For instance, the H₂ evolution rate of 1.25 mmol/h is much smaller than the reported 9.95 mmol/h in ref. 15. The observed maximum power density of 46.3 mW/cm² as a direct hydrazine fuel cell is actually among the lowest in this field! Several hundreds of milliwatts are quite easy to achieve.

5. The possible contribution of Ni foam should be noticed. The authors should be aware that nickel compounds such as Ni₂P show very good electrocatalytic activity in this system.

6. A recent benchmark (Nature Commun. 2019, 10, 1-9, Article number: 4514) related to hydrazine electrooxidation should be cited.

7. Many typos exist, such as "the mixer of H₂ and O₂", "a ultrasml", ...

Reviewer #3 (Remarks to the Author):

The authors proposed a facile two-step strategy to synthesize P,W-codoped cobalt nitride nanowire array material. They characterized the structure and morphology by a number of techniques such as TEM, SEM, XPS, and XRD. They further demonstrated the potential application as bifunctional electrocatalyst for hydrazine oxidation and hydrogen evolution. The catalytic mechanism has been studied by density function theory calculation. This work is publishable in Nature Communicaiotn after addressing the following concerns:

(1) In Figure 5a, the comparison is not fair because of different thermodynamic potentials of OWS (1.23 V) and OH₂S (-0.33 V). I suggest the comparison of faraday efficiencies and overpotentials.

(2) A table should be given to compare the activity of HzOR and HER with the state-of-the-art literature.

(3) What is the efficiency from hydrazine to hydrogen for self-powered H₂ production system? The authors should compare it with other hydrogen generation systems.

(4) The authors described that their catalyst has a "Pt-like" activity. They may need to compare the activity of HER with commercial Pt/C. Further, DFT calculation of HER free energy for Pt should also be given for comparison.

(5) Hydrazine is a highly toxic chemical. Therefore, the authors should discuss further the potential problems and challenges for the large-scale applications of as-proposed self-powered H₂ production system.

Response to Reviewers' Comments

Reviewer #1 (Remarks to the Author):

This manuscript reports the use of P and W co-doped Co_3N for hydrogen evolution and hydrazine oxidation. The catalyst shows good performance. The analysis of the manuscript needs significant improvement, especially for providing the physical insights why P and W co-doped yields good catalysts. I would recommend publication of the manuscript with major revision.

Our response: We appreciate the reviewer for the constructive suggestions on our work. Your valuable suggestions are very helpful for us to further improve the quality of the manuscript. We have studied your comments carefully and conducted additional experiments and analyses accordingly to revise our manuscript.

My detailed comments are as follows:

1. Only P and W co-doping are reported. Have the authors tried only P doped or only W doped samples? That would provide deep insights in the role of co-doping.

Our response: We thank the reviewer for the constructive suggestion, which is very helpful for us to further improve the quality of the manuscript. According to your suggestion, we have synthesized only W doped sample (denoted as W- Co_3N NWA/NF), which was prepared by the same synthesis route except replacing PW_{12} by sodium tungstate dihydrate with the same added amount. Besides, the only P doped sample (denoted as P- Co_3N NWA/NF) was synthesized by the conventional vapor phosphorization method at 300 °C under Ar atmosphere using a spot of sodium hypophosphite as the doping agent (5 mg sodium hypophosphite and a piece of as-prepared Co_3N NWA/NF-2 cm*3 cm). The XRD, XPS and EDS results confirmed the successful doping of W or P while the host phase of Co_3N was maintained. In our experiment, the P or W doped samples exhibited improved HzOR performance compared with that of pure Co_3N NWA/NF. However, their HzOR performance were still inferior compared to that of PW- Co_3N

NWA/NF. Specifically, W-Co₃N NWA/NF required working potentials of -48, 65 and 239 mV to achieve anodic current density of 10, 200 and 600 mA cm⁻². Meanwhile, to reach anodic current density of 10, 200 and 600 mA cm⁻², P-Co₃N NWA/NF needed working potentials of -47, 67 and 191 mV, while the corresponding values for PW-Co₃N NWA/NF were only -55, 27 and 127 mV, respectively. As for HER, PW-Co₃N NWA/NF still owned the best HER activity among all the doped samples. These results indicated that dual doping of P and W is able to improve the electrochemical activity more effectively than the solo doping of P or W. The related material characterizations and electrochemical performance measurements have also been added to Main Text and Supplementary Information part of the revised manuscript. Please see Fig. 4a-b, Fig. 5a-b, Supplementary Fig. 10-16 and Supplementary Fig. 19-20 in the revised manuscript, which are also provided below for your convenience. We also added the related discussions in the main text of the revised manuscript (Lines 7-10, 14-15 and 20, Page 13; Lines 1-2 and 22-23, Page 15; Lines 10-11, Page 16; Lines 19-22, Page 25; Line 1, Page 26).

Supplementary Figure 10. The typical FESEM images and EDS spectrum of W-Co₃N NWA/NF. (a-c) SEM images with different magnifications and (d) EDS spectrum.

Supplementary Figure 11. The XRD pattern of W-Co₃N NWA/NF performed on the powders scratched from Ni foam.

Supplementary Figure 12. The XPS spectra of W-Co₃N NWA/NF. (a) survey spectrum; high resolution spectra of (b) Co 2p, (c) N 1s and (d) W 4f.

Supplementary Figure 13. SEM images and EDS spectrum of P-Co₃N NWA/NF. (a-c) SEM images and (d) EDS spectrum.

Supplementary Figure 14. The XRD pattern of P-Co₃N NWA/NF performed on the powders scratched from Ni foam.

Supplementary Figure 15. The XPS spectra of P-Co₃N NWA/NF. (a) survey spectrum; high resolution spectra of (b) Co 2p, (c) N 1s and (d) P 2p.

Figure 4. Electrocatalytic activity towards HzOR in 1.0 M KOH/0.1 M N₂H₄ electrolyte. (a) Polarization curves and (b) corresponding Tafel plots of PW-Co₃N NWA/NF, W-Co₃N NWA/NF, P-Co₃N NWA/NF, Co₃N NWA/NF, bare Ni foam, PW-Co-precursor and Pt/C; (c) LSV curves of PW-Co₃N NWA/NF with different concentrations of hydrazine, (d) LSV curves of PW-Co₃N NWA/NF electrodes at different scan rates, and the inset is the corresponding current density at 50 mV vs. RHE for different scan rates, (e) Polarization curves of PW-Co₃N NWA/NF electrode after successive CV test; (f) The chronoamperometric test recorded at working potential of -40 mV.

Supplementary Figure 16. Nyquist plots of PW-Co₃N NWA/NF, W-Co₃N NWA/NF, P-Co₃N NWA/NF, Co₃N NWA/NF, Pt/C, Ni foam, PW-Co-precursor and Pt/C obtained at potential of -50 mV for HzOR. The inset is the enlarged view of Nyquist plots of PW-Co₃N NWA/NF, W-Co₃N NWA/NF, P-Co₃N NWA/NF, Co₃N NWA/NF and Pt/C.

Figure 5. Electrocatalytic activity towards HER in 1.0 M KOH electrolyte. (a) Polarization curves of PW-Co₃N NWA/NF, W-Co₃N NWA/NF, P-Co₃N NWA/NF, Co₃N NWA/NF, bare Ni foam, PW-Co-precursor, and Pt/C towards HER, (b) The corresponding Tafel plots derived from (a), (c) Polarization curves of PW-Co₃N NWA/NF before and after CV testing of 1000 and 5000 cycles. The inset is corresponding Nyquist plots. (d) The chronoamperometric test recorded at overpotential of 92 mV.

Supplementary Figure 19. C_{dl} values and cyclic voltammogram curves. (a) C_{dl} values of different materials; CV curves of (b) PW-Co₃N NWA/NF, (c) W-Co₃N NWA/NF, (d) P-Co₃N NWA/NF, (e) Co₃N NWA/NF, (f) Ni foam, (g) PW-Co-precursor and (h) Pt/C in the double layer capacitive region at the scan rates from 10 mV to 100 mV s^{-1} .

Supplementary Figure 20. Nyquist plots of PW-Co₃N NWA/NF, W-Co₃N NWA/NF, P-Co₃N NWA/NF, Co₃N NWA/NF, Ni foam, PW-Co-precursor and Pt/C obtained at overpotential of 100 mV for HER. The inset is the enlarged view of Nyquist plots of PW-Co₃N NWA/NF, W-Co₃N NWA/NF, P-Co₃N NWA/NF, Co₃N NWA/NF, Ni foam, PW-Co-precursor and Pt/C.

2. All the results were obtained using related materials (PW co-precursor, Co₃N, Ni foam) to the current co-doped catalyst, it would be good to use some other known catalysts such as Pt to provide an extra external reference.

Our response: We really appreciate the reviewer for the professional suggestion. According to your suggestion, the HzOR and HER catalytic performance of commercial Pt/C (20 wt.%) with the identical mass loading of 2 mg cm⁻² to that of PW-Co₃N NWA/NF sample have been investigated as comparison and the results were also added to Fig. 4a-b, Fig. 5a-b, Supplementary Fig. 16 and Supplementary Fig. 19-20 of the revised manuscript, which were also provided below for your convenience. As indicated, our PW-Co₃N NWA/NF exhibited much better electrocatalytic activity for HzOR compared with that of Pt/C in 1.0 M KOH/0.1 M N₂H₄. Specifically, Pt/C required working potentials of 14 and 223 mV to achieve anodic current density of 10 and 200 mA cm⁻². For HER, the overpotential of PW-Co₃N NWA/NF is 9 mV larger than that of Pt/C (32 mV) at 10 mA cm⁻², while the Tafel slope of PW-Co₃N NWA/NF is 40 mV dec⁻¹, which is also comparable to commercial Pt/C (31 mV dec⁻¹). The electrochemical performance of known benchmark Pt/C as an extra external reference further proved the exciting HzOR and HER performance of PW-Co₃N NWA/NF. The related discussions are also added to

the main text of the revised manuscript. Please see Lines 10 and 21, Page 13; Lines 3 and 22-23, Page 15; Line 13, Page 16.

Figure 4. Electrochemical activity towards HzOR in 1.0 M KOH/0.1 M N_2H_4 electrolyte. (a) Polarization curves and (b) corresponding Tafel plots of PW- Co_3N NWA/NF, W- Co_3N NWA/NF, P- Co_3N NWA/NF, Co_3N NWA/NF, bare Ni foam, PW-Co-precursor and Pt/C; (c) LSV curves of PW- Co_3N NWA/NF with different concentrations of hydrazine, (d) LSV curves of PW- Co_3N NWA/NF electrodes at different scan rates, and the inset is the corresponding current density at 50 mV vs. RHE for different scan rates, (e) Polarization curves of PW- Co_3N NWA/NF electrode after successive CV test; (f) The chronoamperometric test recorded at working potential of -40 mV.

Supplementary Figure 16. Nyquist plots of PW-Co₃N NWA/NF, W-Co₃N NWA/NF, P-Co₃N NWA/NF, Co₃N NWA/NF, Pt/C, Ni foam, PW-Co-precursor and Pt/C obtained at potential of -50 mV for HzOR. The inset is the enlarged view of Nyquist plots of PW-Co₃N NWA/NF, W-Co₃N NWA/NF, P-Co₃N NWA/NF, Co₃N NWA/NF and Pt/C.

Figure 5. Electrocatalytic activity towards HER in 1.0 M KOH electrolyte: (a) Polarization curves of PW-Co₃N NWA/NF, W-Co₃N NWA/NF, P-Co₃N NWA/NF, Co₃N NWA/NF, Ni foam, PW-Co-precursor and Pt/C towards HER, (b) The corresponding Tafel plots derived from (a), (c) Polarization curves of PW-Co₃N NWA/NF before and after CV testing of 1000 and 5000 cycles. The inset is corresponding Nyquist plots. (d) The chronoamperometric test recorded at overpotential of 92 mV.

Supplementary Figure 19. C_{dl} values and cyclic voltammogram curves. (a) C_{dl} values of different materials; CV curves of (b) PW-Co₃N NWA/NF, (c) W-Co₃N NWA/NF, (d) P-Co₃N NWA/NF, (e) Co₃N NWA/NF, (f) Ni foam, (g) PW-Co-precursor and (h) Pt/C in the double layer capacitive region at the scan rates from 10 mV to 100 mV s^{-1} .

Supplementary Figure 20. Nyquist plots of PW-Co₃N NWA/NF, W-Co₃N NWA/NF, P-Co₃N NWA/NF, Co₃N NWA/NF, Ni foam, PW-Co-precursor and Pt/C obtained at overpotential of 100 mV for HER. The inset is the enlarged view of Nyquist plots of PW-Co₃N NWA/NF, W-Co₃N NWA/NF, P-Co₃N NWA/NF, Co₃N NWA/NF, Ni foam, PW-Co-precursor and Pt/C.

3. On page 13, can the authors comment on why the electrochemical surface area for PW-Co₃N is so much bigger or have more number of accessible active sites?

Our response: We thank the reviewer for the good question. Indeed, the electrochemical surface area of PW-Co₃N NWA/NF is nearly two times larger than that of pure Co₃N NWA/NF. The possible reasons for the much enhanced electrochemical surface area can be illustrated as follows: Firstly, the P/W co-doping could effectively modulate the electronic structure of Co₃N confirmed by the experimental data based on the XPS and XANES results (Fig. 2 and Fig. 3 in the revised manuscript) together with the added DFT calculation on charge density difference analysis (Fig. 8h-i in the revised manuscript). It can be concluded that the electron density around Co and N atoms can be effectively modulated after doping, which could contribute to the catalytic performance of PW-Co₃N. Secondly, the hydrogen absorption kinetics can be modified to more thermoneutral and the H₂O adsorption is further facilitated after doping according to the DFT calculation (Fig. 8e and g in the revised manuscript), which can further enhance the catalytic activity (*Nat. Commun.* 2019, **10**, 1743; *Angew. Chem. Int. Ed.* 2019, **58**, 11903-11909). In short, the much enhanced electrochemical surface area can be attributed to the doping induced electronic structure modulation and hydrogen absorption kinetics manipulation, which have also

been observed in previous literatures (*Adv. Mater.* 2019, **31**, 1901174; *Angew. Chem. Int. Ed.* 2018, **57**, 5076–5080; *Adv. Funct. Mater.* 2017, **27**, 1704169).

4. The authors cited “best reported values” in many places. It would be much more informative if they listed a few concrete values and systems.

Our response: We thank the reviewer for the constructive suggestion, which is very important for us to further improve the quality of our manuscript. According to your suggestion, we added two Tables (Supplementary Table 2 and Table 3) to compare the HzOR and HER performance of PW-Co₃N NWA/NF and recently reported literatures with more detailed information such as materials system, detailed overpotential/working potential values and electrolyte compositions. The relevant discussions have also been added and highlighted in the main text of the revised manuscript (Lines 16-18, Page 13; Lines 16-19, Page 15). We also provided the Tables below for your convenience.

Supplementary Table 2. Comparison of the electrocatalytic activities of PW-Co₃N NWA/NF with other reported materials for HzOR.

Materials	electrolyte	J (mA cm ⁻²)	Potential (mV)	Reference
		10	-55	
PW-Co ₃ N NWA/NF	1.0 M KOH+0.1 M N ₂ H ₄	50	-29	This work
		200	27	
Co ₃ Ta/C	3.0 M KOH+0.5 M N ₂ H ₄	25.2	60	Nat. Commun. 10 , 4514 (2019).

Cu₁Ni₂-N	1.0 M KOH+0.5 M N ₂ H ₄	10	0.5	Adv. Energy Mater. 9 , 1900390 (2019).
Ni_xP/NF	1.0 M NaOH+0.1 M N ₂ H ₄	172	100	Appl. Catal. B: Environ. 241 , 292-298 (2019).
Rh/N-CBs	1.0 M KOH+0.05 M N ₂ H ₄	10	72	ACS Appl. Mater. Interfaces 11 ,35039-3504 (2019).
Fe-CoS₂	1.0 M KOH+0.1 M N ₂ H ₄	100	129	Nat. Commun. 9 , 4365 (2018).
CoSe₂	1.0 M KOH+0.5 M N ₂ H ₄	10	-17	Angew. Chem. Int. Ed. 57 , 7649-7653 (2018).
Ni₃S₂/NF	1.0 M KOH+0.2 M N ₂ H ₄	100	415	J. Mater. Chem. A 6 , 19201- 19209 (2018)
Ni₂P/NF	1.0 M KOH+0.5 M N ₂ H ₄	50	-25	Angew. Chem. Int. Ed. 56 , 842-846 (2017).
Ni-NSA	3.0 M KOH+1.0 M N ₂ H ₄	227.6	250	Angew. Chem. Int. Ed. 55 , 693-697 (2016).
NiZn	1.0 M KOH+1.0 M N ₂ H ₄	320	600	Angew. Chem. Int. Ed. 53 , 10336-0339 (2014).

Supplementary Table 3. Comparison of the electrocatalytic activities of PW-Co₃N NWA/NF with recently reported transition metal nitride for HER in 1.0 M KOH.

Materials	η_{10} (mV)	Tafel slope (mV dec⁻¹)	Reference
PW-Co₃N NWA/NF	41	40	This work

NiMoN/NF	56	45.6	Nat. Commun. 10 , 5106 (2019).
NiCoN/C nanocages	103	-	Adv. Mater. 31 , 1805541 (2019).
Ni ₃ N/C	64	48	Angew. Chem. Int. Ed. 58 , 1-6 (2019).
Co ₂ N/Co/CF	12	41.6	ACS Energy Lett. 4 , 1594-1601 (2019).
V-Co ₄ N/NF	37	44	Angew. Chem. Int. Ed. 57 , 5076-5080 (2018).
Co-Ni ₃ N	194	156	Adv. Mater. 30 , 1705516 (2018).

5. On page 16, it mentioned “energetically favorable to replace surface Co atoms ...”. Please provide those energetic data.

Our response: We thank the reviewer for the good suggestion. According to your suggestion, we provided the energetic data of two different models as Supplementary Fig. 26 of the revised manuscript, which is also provided below for your convenience. As can be seen, the total energy of system after surface substitution of 4 Co with 4 W atoms at Co₃N (001) plane is -328.12 eV with no visible structural distortion, which is consistent with the XRD results before and after P/W doping. Contrastively, there will be serious structural distortion for the system with 4 W doped at the subsurface of Co₃N (001) planes where the N atoms of second layer totally deviate from its original position, as shown in Supplementary Fig. 26b. This is not reasonable since the crystal structure can be well maintained after W doping according to the XRD results, although the total energy (-333.76 eV) is lower compared to the system with surface doping model. Based on these results, we adopt the optimal structure of surface doping model during our calculation.

Supplementary Figure 26. Model simulation of 4 W atoms and 1 P atom substituted Co_3N .

(a) Model simulation of 4 W atoms substituted surface Co sites and (b) substituted subsurface Co sites, before (left) and after (right) geometric optimization; The balls in yellow, pink, green and purple represent Co, W, P and N atoms, respectively.

6. While the hydrogen adsorption free energy becomes less negative in the co-doped case, it is still far from the conventional optimal value of 0. Even pure Ni has free energy of ~ -0.3 eV. It is not obvious from Fig. 7b, why the catalyst has favorable HER activity.

Our response: We thank the reviewer for the professional question. In order to further understand the possible reasons for the favorable HER activity of the PW- Co_3N NWA/NF, we conducted the extra DFT calculations on d-band center analysis, the water absorption energy and charge density difference analysis. The related results are added to Fig. 8 in the revised manuscript, which are also provided below for your convenience. As indicated, the d-band center of pure Co_3N (-1.41 eV) notably shifts down to -1.54 eV after P/W doping (Fig. 8f), which clearly demonstrates that hydrogen desorption is promoted besides the decreasing of hydrogen absorption energy. In addition, the adsorption energy of H_2O ($\Delta E_{\text{H}_2\text{O}}$) on Co_3N (001) and PW-

Co_3N (001) are -0.31 and -0.56 eV (Fig. 8g), respectively, which indicates that the adsorption of H_2O molecule is energetically more favorable on the PW- Co_3N surface. These results suggest that the P/W doping can not only manipulate the hydrogen absorption energy, but also tailor the d-band center and facilitate the absorption of H_2O , which can synergistically contribute to the HER activity. On the other hand, the charge density difference analysis (Fig. 8h, i) shows that the charge redistribution is dominantly restricted on P/W and their nearest neighboring Co/N, which could potentially enhance the catalytic activity of PW- Co_3N compared to pure Co_3N . Moreover, the integrated electrode architecture by direct growth of active component on Ni foam could largely facilitate the electron transfer behavior, which could be another important factor that our PW- Co_3N NWA/NF exhibits favorable HER activity. The related discussions have also been added to Line 23, Page 21; Lines 11-13, Page 22 and Lines 1-12, Page 23 of the revised manuscript.

Figure 8. DFT calculated profiles of free energy: Top- and side- view of atomic structure models for (a, c) Co₃N; (b, d) PW-Co₃N; (e) The free energy diagram of HER at the equilibrium potential for Co₃N and PW-Co₃N, H* denotes that intermediate adsorbed hydrogen; (f) The d band of density of states (DOS) of Co₃N and PW-Co₃N; (g) Water adsorption energy on Co₃N and PW-Co₃N; Top-(h) and side-(i) view of charge density difference analysis for PW-Co₃N with the cyan region representing charge depletion and the yellow region representing charge accumulation. The isosurface value is 0.012 eÅ⁻³; (j) The free energy profiles of HzOR on the Co₃N and PW-Co₃N surfaces. The inset in (j) are the most stable configurations of the each adsorbed intermediate on the Co site.

7. N in figure 7f is hard to see. Please consider revise the color scheme.

Our response: We really appreciate the reviewer for the good suggestion. According to your suggestion, we have modified the color scheme of Fig. 8 in the revised manuscript.

8. I would recommend performing some charge analysis, in the form of calculating Bader charges or obtaining charge density difference plot, to understand what is the impact of co-doping on Co_3N .

Our response: We thank the reviewer for the good suggestion. According to your suggestion, we performed charge density difference plot of $\text{PW-Co}_3\text{N}$, as shown below. As can be seen, the charge redistribution is dominantly restricted on P/W and their nearest neighboring Co/N, and the doping W and nearby Co possess charge depletion while P and the neighboring N possess charge accumulation. This electron localization behavior upon doping could contribute to the enhanced catalytic activity of $\text{PW-Co}_3\text{N}$. The related results are added to Fig. 8h-i in the revised manuscript. The related discussions have also been added to the revised manuscript (Lines 8-12, Page 23).

9. Significant justification is needed for the co-doping calculation model. Why the Co-termination unit cell is adopted? What is the current W coverage? Why using 4 W but only one P? Why only surface substitution is used, but not subsurface ones?

Our response: We really appreciate the reviewer's professional questions. We are sorry that we didn't provide clear instruction about the reason why we adopted the Co-termination unit cell as the model for DFT calculation. In fact, when we performed the DFT calculation, these two different types of unit cells, i. e. Co-terminated and N-terminated were both tried. As shown in Supplementary Fig. 27, the distance between N_2H_4 molecular and the nearest neighboring N atom is 5.39 Å, which is much larger than the distance of 1.98 Å between N_2H_4 and Co atom. This means that the N atom terminated unit cell is not favorable for N_2H_4 adsorption, which is inconsistent with our experimental results. Therefore, we adopted the Co-termination unit cell as the model for DFT calculation. The calculation results have been added as Supplementary Fig. 27 and the related discussions were added to Lines 17-22, Page 23 in the revised manuscript.

Supplementary Figure 27. Models simulating adsorption of N_2H_4 molecule on the (a) N atom terminal and (b) Co atom terminal of $Co_3N(001)$, respectively, before (left) and after (right) geometric optimization. The balls in yellow and purple represent Co and N atoms, respectively.

The W coverage is confirmed according to the atomic ratios from EDS and ICP-AES results, which is provided below. In experiment, the P: W: Co is about 1: 3.50: 36.9 calculated from EDS and ICP-AES measurements. In DFT calculation, the atom number for P, W and Co is 1, 4 and 32 in order to be close to the experimental results. The related data were added as Supplementary Fig. 3 and the related discussions were provided in Lines 15-17, Page 8 of the revised manuscript.

Regarding the reason to adopt the surface substitution model during the DFT calculation, we confirmed this according to the calculated energetic data of two different models (See Figure S26 below). As can be seen from Supplementary Fig. 26a, the total energy of system after surface substitution of 4 Co with 4 W atoms at Co_3N (001) plane is -328.12 eV with no visible structural distortion, which is consistent with the XRD results before and after P/W doping. Contrastively, there will be serious structural distortion for the system with 4 W doped at the subsurface of Co_3N (001) planes where the N atoms of second layer totally deviate from its original position, as shown in Supplementary Fig. 26b. This is not reasonable since the crystal structure can be well maintained after W doping according to the XRD results, although the total energy (-333.76 eV) is slightly lower to the system with surface doping model. Based on these results, we adopt the optimal structure of surface doping model during our calculation.

Supplementary Figure 3. EDS spectrum of PW- Co_3N NWA/NF.

Supplementary Figure 26. Model simulation of 4 W atoms and 1 P atom substituted Co_3N .

(a) Model simulation of 4 W atoms substituted surface Co sites and (b) substituted subsurface Co sites, before (left) and after (right) geometric optimization; The balls in yellow, pink, green and purple represent Co, W, P and N atoms, respectively.

Reviewer #2 (Remarks to the Author):

This manuscript by Liu and co-workers described P, W co-doped cobalt nitride nanowire arrays for catalytic hydrazine oxidation and hydrogen evolution. In general, the resulted nanocomposites demonstrated quite impressive activities for both reactions, which allowed the integration of a direct hydrazine fuel cell with a hydrazine splitting electrolyzer for self-powered H₂ production. However, the manuscript contains substantial unproven or unnecessary claims and its overall quality does not warrant its suitability for publication in leading journals such as Nature Commun.

Our response: Thank you very much for your precious time on reviewing our manuscript. We have carefully studied your comments and conducted a series of additional experiments with related discussions according to your suggestions. Also, we tried our best to answer your questions point-to-point, which are provided below.

1. The fabrication method is quite routine, and there are many hundreds of similar works to decorate various electrochemically active materials onto Ni foams for various electrocatalytic reasons such as OER, HER, CO₂RR, N₂RR, and HzOR. And this work is no exception.

Our response: We thank the reviewer for the comment.

We would like to highlight the novelty and significance of our work in the following aspects:

(i) Indeed, it is true that plenty of works have been reported regarding the self-supported electrode by direct growth of active materials onto Ni foam or other conductive substrates including carbon cloth, Ti and Cu foils and so on (*Adv. Mater.* 2019, 1806326; *Adv. Mater.* 2019, **31**, 1901174; *Adv. Mater.* 2019, **31**, 1903909; *Nature Commun.* 2018, **9**, 924; *Nature Commun.* 2018, **9**, 4531; *Adv. Mater.* 2016, **28**, 215-230; *Nature Commun.* 2015, **6**, 6616; *Angew. Chem. Int. Ed.* 2015, **54**, 9351-9355). However, to obtain smart materials using the relatively facile synthesis strategy with unique advantages of requiring low-cost equipment and chemicals, but

exhibiting superior performance and providing sufficient scientific contribution on related areas, is one of the significant targets that we are sedulously pursuing as a researcher on materials science.

(ii) Moreover, although the self-supported electrodes based on Ni foam have been widely reported, most of the literatures are focusing on HER and OER, while there are very limited literatures regarding the transition metal nitrides for HzOR. To the best of our knowledge, our work regarding the P/W co-doped Co_3N on Ni foam is the first work to demonstrate that the heteroatom doping can largely enhance the HzOR catalytic activity of transition metal nitrides, and more importantly, can simultaneously improve the HER activity. The results in our work provide a promising strategy for transition metal nitrides to simultaneously modify the HzOR and HER activity, which is significant for both fundamental research and extended application for H_2 production.

(iii) On the other hand, it is one of important standards to evaluate the significance of the research work that if the performance of the materials was further improved compared with the state-of-the-art literatures. In our work, we are confident that the HzOR catalytic activity of PW- Co_3N NWA/NF is superior compared to recent state-of-the-art literatures about different material systems. Please see the added Supplementary Table 2 for the detailed information, which is also provided below for your convenience. In addition, as a demonstration for potential applications, we built a “self-powered” H_2 production system using DHzFC driven OHzS based on PW- Co_3N NWA/NF for both HzOR and HER catalysts, where a decent H_2 production rate of 1.25 mmol h^{-1} can be achieved. It should be emphasized that the reported H_2 production rate is achieved with the DHzFC working at room temperature with electrolyte composed of 1 M KOH/0.5 M hydrazine and with the two-electrode OHzS electrolyzer working with the 1 M KOH/0.5 M hydrazine electrolyte. The power density of the DHzFC is highly related with the working temperature and the H_2 production rate is also dependent on the concentration of hydrazine in the

electrolyte (*Adv. Mater.* 2015, **27**, 2361-2366). The reported performance in our work regarding the maximum power density and the H₂ production rate obtained in the self-power system are superior under the same or close conditions of working temperature and electrolyte components.

More detailed explanations are provided regarding the response to the reviewer's comment (4).

(iv) Last but not the least, it is meaningful to disclose the possible origin of the enhanced catalytic activity after doping by density functional theory (DFT) calculations. In our work, the calculation results indicate that the P/W doping can not only manipulate the dehydrogenation kinetics for HzOR, but also simultaneously modulate the hydrogen absorption energy. Moreover, in the revised manuscript, we further performed the charge density difference analysis, combined with experimental results including XPS and XANES characterizations (Fig. 2), which show that the P/W doping can also modulate electronic structure of Co₃N. Additionally, the added FT and WT of EXAFS characterizations (Fig. 3) in the revised manuscript verify that the doped W substitutes Co site and bonds with P and N. In short, these results can provide a meaningful reference for future research on related material systems.

Supplementary Table 2. Comparison of the electrocatalytic activities of PW-Co₃N NWA/NF with other reported materials for HzOR.

Materials	electrolyte	J (mA cm ⁻²)	Potential (mV)	Reference
		10	-55	
PW-Co₃N NWA/NF	1.0 M KOH+0.1 M N ₂ H ₄	50	-29	This work
		200	27	

Co₃Ta/C	3.0 M KOH+0.5 M N ₂ H ₄	25.2	60	Nat. Commun. 10 , 4514 (2019).
Cu₁Ni₂-N	1.0 M KOH+0.5 M N ₂ H ₄	10	0.5	Adv. Energy Mater. 9 , 1900390 (2019).
Ni_xP/NF	1.0 M NaOH+0.1 M N ₂ H ₄	172	100	Appl. Catal. B: Environ. 241 , 292-298 (2019).
Rh/N-CBs	1.0 M KOH+0.05 M N ₂ H ₄	10	72	ACS Appl. Mater. Interfaces 11 ,35039-3504 (2019).
Fe-CoS₂	1.0 M KOH+0.1 M N ₂ H ₄	100	129	Nat. Commun. 9 , 4365 (2018).
CoSe₂	1.0 M KOH+0.5 M N ₂ H ₄	10	-17	Angew. Chem. Int. Ed. 57 , 7649-7653 (2018).
Ni₃S₂/NF	1.0 M KOH+0.2 M N ₂ H ₄	100	415	J. Mater. Chem. A 6 , 19201- 19209 (2018)
Ni₂P/NF	1.0 M KOH+0.5 M N ₂ H ₄	50	-25	Angew. Chem. Int. Ed. 56 , 842-846 (2017).
Ni-NSA	3.0 M KOH+1.0 M N ₂ H ₄	227.6	250	Angew. Chem. Int. Ed. 55 , 693-697 (2016).
NiZn	1.0 M KOH+1.0 M N ₂ H ₄	320	600	Angew. Chem. Int. Ed. 53 , 10336-0339 (2014).

2. The structure characterization is very poor. There is no discussion of how and where P and W species locate within the Co₃N lattice. HRTEM imaging and X-ray absorption seem necessary to address this critical issue. The authors did not even provide the chemical composition of their materials system. Without all these key information, what is the reliability for their DFT models?

Our response: We thank the reviewer for the important questions. In order to tackle these

questions, we have added aberration corrected HAADF-STEM measurements for PW-Co₃N NWA/NF with atomic resolution and XAS characterizations (both XANES and EXAFS) in order to further strengthen the structure characterizations.

The atomically resolved HAADF-STEM images are added as Fig. 1h-k in the revised manuscript, which is also provided below for your convenience. As can be seen in Fig. 1h and i, the W atoms can be seen as bright dots in Co₃N lattice due to Z-contrast in HAADF-STEM image, since the atomic number of W is significantly larger than that of Co. However, the brightness of W atoms is not so obvious due to the relatively thick Co₃N substrate, especially when viewing along the zone axis. We further tilted the TEM holder to make the specimen off the zone axis, so that the lattice of Co₃N cannot be clearly seen, and then the brightness of W showed up (Fig. 1j).

The X-ray absorption (XAS) results were added to Fig. 2 and Fig. 3 in the revised manuscript, which are also provided below for your convenience. As indicated, the XANES results for Co L-edge and N K-edge show a positive shift for the Co L₃/L₂ peaks and a negative shift for N K-edge peaks after P/W doping, which is consistent with the XPS results and can further verify the existed charge transfer due to the P/W doping. This phenomenon is also consistent with the DFT calculation results on charge density difference analysis (Provided as Fig. 8h in the revised manuscript). Moreover, observed from FT of the Co K edge EXAFS spectra (Fig. 3b), the intensity of Co-Co peak decreases obviously in PW-Co₃N, suggesting the success doping of P/W. Besides, the FT of W L₃-edge EXAFS spectra (Fig. 3f) declare completely different coordination environment of W in PW-Co₃N compared with W foil and WO₃, which further proves that the W is successfully doped into Co₃N. We also performed FT-EXAFS fitting to get insight on the coordination environment of doping W in PW-Co₃N, as shown in Fig. 3g, the fitting results are summarized in Supplementary Table 1, which are also provided below for your convince. The fitting results indicate clearly W–N bond, W–P bond and

W-Co bond, which confirms that the doped W substitutes Co sites and bonds with P and N. Moreover, the WT of W L_3 -edge contour plots of PW-Co₃N (Fig. 3h) exhibit the typical W-Co bond (centered about 7.5 Å⁻¹) and Co-N bond (centered about 6.0 Å⁻¹), which further supports that W substitutes Co site. The related discussions have also been added to the main text of the manuscript. Please see Lines 7-15, Page 8; Page 17-23, Page 10; Lines 1-23, Page 11; Lines 8-14, Page 12 of the revised manuscript.

Figure 1. Morphological and structural characterization of PW-Co₃N NWA/NF: (a) Schematic illustration of the formation process; (b, c, d) FESEM and (e) TEM images; (f) HRTEM image; (g) HAADF-STEM image and corresponding elemental mapping results; (h) aberration corrected HAADF-STEM image; (i) the enlarged picture from (h); (j) atomically resolved HAADF-STEM image; (k) the corresponding FFT image of (h).

Figure 2. High resolution XPS spectra and soft XANES of PW-Co₃N NWA/NF and Co₃N NWA/NF: high resolution XPS spectra of (a) Co 2p, (b) N 1s, (c) W 4f, and (d) P 2p; soft XANES of (e) Co L-edge spectra; (f) N K-edge spectra.

Figure 3. XANES and EXAFS spectra of PW-Co₃N NWA/NF and Co₃N NWA/NF: (a) The normalized Co K-edge spectra and (b) FT of the Co K-edge EXAFS of PW-Co₃N, Co₃N, Co foil and CoO (line: raw data, scatter: fit); WT of the Co K-edge EXAFS contour plots of (c) Co₃N and (d) PW-Co₃N; (e) The normalized W L₃-edge spectra and (f) FT of the W L₃-edge EXAFS of PW-Co₃N, W foil and WO₃ (line: raw data, scatter: fit); (g) The FT of the W L₃-edge EXAFS fitting curves of PW-Co₃N; (h) WT of the W L₃-edge EXAFS contour plots of PW-Co₃N.

Figure 8. DFT calculated profiles of free energy: Top- and side- view of atomic structure models for (a, c) Co₃N; (b, d) PW-Co₃N; (e) The free energy diagram of HER at the equilibrium potential for Co₃N and PW-Co₃N, H* denotes that intermediate adsorbed hydrogen; (f) The d band of density of states (DOS) of Co₃N and PW-Co₃N; (g) Water adsorption energy on Co₃N and PW-Co₃N; Top-(h) and side-(i) view of charge density difference analysis for PW-Co₃N with the cyan region representing charge depletion and the yellow region representing charge accumulation. The isosurface value is 0.012 eÅ⁻³; (j) The free energy profiles of HzOR on the Co₃N and PW-Co₃N surfaces. The inset in (j) are the most stable configurations of the each adsorbed intermediate on the Co site.

Supplementary Table 1. EXAFS fitting parameters at the Co K-edge and W L₃-edge for various samples. $S_0^2 = 0.776(\text{Co}), 0.859(\text{W})$.

Sample	Shell	N^a	$R(\text{\AA})^b$	$\sigma^2(\text{\AA}^2)^c$	$\Delta E_0(\text{eV})^d$	R factor
Co K-edge						
Co foil	Co-Co	12	2.49	0.0063	7.2	0.0004
CoO	Co-O	6.0	2.06	0.0101	-3.6	0.0004
	Co-Co	12.0	2.99	0.0095		
Co₃N	Co-N	2.1	1.88	0.0019	-0.6	0.0007
	Co-Co	11.6	2.67	0.0073		
PW-Co₃N	Co-N	2.0	1.85	0.0021	-0.9	0.0065
	Co-Co	12.9	2.67	0.0119		
W L₃-edge						
W foil	W-W	8	2.74	0.0027	5.8	0.0040
	W-W	6	3.16	0.0027		

	W-O	4.0	1.76	0.0035		
	W-O	2.0	2.16	0.0018		
					8.7	0.0082
WO₃	W-O	2.1	3.22	0.0034		
	W-W	7.9	3.66	0.0062		
	W-N	1.7	1.79	0.0015		
	W-P	1.2	2.33	0.0048		
PW-Co₃N					1.0	0.0015
	W-Co	4.4	3.09	0.0096		
	W-Co	4.5	3.34	0.0096		

^a*N*: coordination numbers; ^b*R*: bond distance; ^c σ^2 : Debye-Waller factors; ^d ΔE_0 : the inner potential correction. *R* factor: goodness of fit. S_0^2 was set to 0.776 for Co and 0.859 for W, according to the experimental EXAFS fit of Co and W foil reference by fixing CN as the known crystallographic value.

Regarding the chemical compositions of the PW-Co₃N NWA/NF sample, we are sorry that this data was missing in the original submission. The chemical composition in our work was confirmed by EDS and Inductive coupled plasma-atomic emission spectrometry (ICP-AES) measurements, where the atomic ratio of P : W: Co is about 1: 3.50: 36.9. The doping level in the DFT calculation is determined according to the experimental results, where the P: W: Co ratio is 1:4:32. The related discussions were provided in Lines 15-17, Page 8 of the revised manuscript.

Supplementary Figure 3. EDS spectrum of PW-Co₃N NWA/NF.

3. The reviewer did not see the particular advantage of so called self-powered H₂ production system, because direct splitting of hydrazine is very straightforward to produce H₂ and N₂ within the scope of heterogeneous catalysis. Besides, this self-powered H₂ production system has been reported by the authors of Ref. 15. So essentially there is nothing new here.

Our response: We thank the reviewer for the comment. In our work, we firstly demonstrate that the hugely enhanced HzOR catalytic activity of transition metal nitrides coupled with simultaneously improved HER activity can be achieved via the heteroatom doping of P/W, which provides a promising strategy to modify the HzOR and HER activity of transition metal nitrides concurrently, as well as has a great significance for both fundamental research and extended application of H₂ production. Meanwhile, the HzOR catalytic activity of PW-Co₃N NWA/NF in our work is superior compared to recent state-of-the-art literatures about different material systems (Please see the added Supplementary Table 2). As a demonstration for potential applications, the “self-powered” H₂ production system using DHzFC driven OHzS based on PW-Co₃N NWA/NF for both HzOR and HER catalysts were built, where a decent H₂ production rate of 1.25 mmol h⁻¹ can be achieved. Moreover, DFT calculations were performed to disclose the possible origin of the enhanced catalytic activity after doping, which indicated that the P/W doping can not only manipulate the dehydrogenation kinetics for HzOR, but also simultaneously modulate the hydrogen absorption energy. Besides, the further performed theoretical charge

density difference analysis, combined with experimental results including XPS and XANES characterizations in our revised manuscript additionally show that the P/W doping can also cause charge redistribution, which is restricted on P/W and their nearest neighboring Co/N, thus modulating electronic structure of Co₃N. Moreover, FT and WT of EXAFS characterizations reveal that doped W substitutes Co site and bond with P and N. Briefly, these results from our work can provide a meaningful reference for future research on related material systems.

As a conclusion, the most important scientific contribution of our work is the first report demonstrating the notable effect of heteroatom doping of transition metal nitride can largely improve its HzOR activity, and simultaneously enhance the HER activity in certain degree. The theoretical DFT calculation combined with systematic experimental data disclose the fundamental origin of the enhanced performance. We are confident that our work regarding the P/W doped Co₃N nanowire array electrode possesses sufficient novelty regarding the reported strategy for simultaneous enhancement for HzOR and HER activity, as well as significant scientific contribution regarding the fundamental understanding over the structure-property relationship based on the combined DFT calculation and experimental data.

4. By claiming "a record high", "outperforms most of the literatures", "is one of the best reported values", "is superior compared to the reported values", "the exciting potential for highly efficient H₂ productions", "possesses benchmark electrocatalytic activity", "is superior compared to state-of-the-art values", the authors should at least provides sufficient literatures for direct comparison (such as a table?). Unfortunately, the authors did not even conduct control experiments using standard electrode materials such as Pt/C, which makes the data less convincing. For instance, the H₂ evolution rate of 1.25 mmol/h is much smaller than the reported 9.95 mmol/h in ref. 15. The observed maximum power density of 46.3 mW/cm² as a direct hydrazine fuel cell is actually among the lowest in this field! Several hundreds of milliwatts are quite easy to achieve.

Our response: We thank the reviewer for the comment. Firstly, it is necessary to clarify that the

maximum power density of the DHzFC is highly related with the working temperature and the H₂ evolution rate in two-electrode OHzS electrolyzer is highly dependent on the hydrazine concentration in the electrolyte. In Ref. 15 of origin manuscript (*Nat. Commun.* 2018, **9**, 4365), the H₂ evolution rate of 9.95 mmol h⁻¹ is obtained with the high hydrazine concentration of 5.6 M in the two-electrode OHzS electrolyzer, powered by DHzFC working under 80 °C with a maximum power density of 125 mW cm⁻². In our work, the H₂ evolution rate is obtained with a much lower hydrazine concentration of 0.5 M in the two-electrode OHzS electrolyzer, powered by DHzFC working at room temperature (~25 °C). When we claimed the performance of our material as “superior or benchmark”, it is for sure that we mean the performance comparison under similar measurement conditions. For example, in the Supplementary Information of Ref. 45 of revised manuscript (*Adv. Mater.* 2015, **27**, 2361-2366), the authors provided the maximum power density of their DHzFC under room temperature in Supplementary Fig.12, which is about 29.1 mW cm⁻². In order to clearly compare our results with recently published literatures regarding the maximum power density of DHzFC under room temperature, we added a table as Supplementary Table 4 in the revised manuscript, which can solidly support our claim regarding the performance of the DHzFC based on our PW-Co₃N NWA/NF as anode catalyst. Moreover, the corresponding H₂ evolution rate under the same hydrazine concentration of 0.5 M in Ref.15 of origin manuscript is about 0.82 mmol h⁻¹ even when powered by the DHzFC working under 80 °C (Supplementary Fig. 27 of Ref. 15 of origin manuscript). It is clear that our PW-Co₃N NWA/NF indeed exhibits better performance compared to these recently published state-of-the-art values for the maximum power density of DHzFC and H₂ evolution rate in the self-powered system under the identical working conditions.

According to your suggestions, we also summarized the recent literatures about HzOR, HER and OHzS system, which are added as Supplementary Table 2-4 in the revised manuscript, respectively, in order to provide direct comparison with sufficient literatures. Besides, the HzOR

and HER performance of commercial Pt/C have been collected. Please see Fig. 4a-b, Fig. 5a-b, Supplementary Fig. 16 and Supplementary Fig. 19-20 in the revised manuscript, which are also provided below for your convenience. For HzOR, the activity of PW-Co₃N NWA/NF notably outperforms that of Pt/C, which requires working potentials of 14 and 223 mV to achieve anodic current density of 10 and 200 mA cm⁻². For HER, the overpotential of PW-Co₃N NWA/NF is 9 mV larger than that of Pt/C (32 mV) at 10 mA cm⁻² with comparable Tafel slopes of 40 mV dec⁻¹ (31 mV dec⁻¹ for Pt/C). The related discussions for the HzOR and HER performance of Pt/C are also added in the main text of the revised manuscript (Lines 10 and 21, Page 13; Lines 3 and 22-23, Page 15; Line 13, Page 16).

Figure 4. Electrocatalytic activity towards HzOR in 1.0 M KOH/0.1 M N₂H₄ electrolyte. (a) Polarization curves and (b) corresponding Tafel plots of PW-Co₃N NWA/NF, W-Co₃N NWA/NF, P-Co₃N NWA/NF, Co₃N NWA/NF, bare Ni foam, PW-Co-precursor and Pt/C; (c) LSV curves of PW-Co₃N NWA/NF with different concentrations of hydrazine, (d) LSV curves of PW-Co₃N NWA/NF electrodes at different scan rates, and the inset is the corresponding current density at 50 mV vs. RHE for different scan rates, (e) Polarization curves of PW-Co₃N NWA/NF electrode after successive CV test; (f) The chronoamperometric test recorded at working potential of -40 mV.

Supplementary Figure 16. Nyquist plots of PW-Co₃N NWA/NF, W-Co₃N NWA/NF, P-Co₃N NWA/NF, Co₃N NWA/NF, Pt/C, Ni foam, PW-Co-precursor and Pt/C obtained at potential of -50 mV for HzOR. The inset is the enlarged view of Nyquist plots of PW-Co₃N NWA/NF, W-Co₃N NWA/NF, P-Co₃N NWA/NF, Co₃N NWA/NF and Pt/C.

Figure 5. Electrocatalytic activity towards HER in 1.0 M KOH electrolyte. (a) Polarization curves of PW-Co₃N NWA/NF, W-Co₃N NWA/NF, P-Co₃N NWA/NF, Co₃N NWA/NF, bare Ni foam, PW-Co-precursor, and Pt/C towards HER, (b) The corresponding Tafel plots derived from (a), (c) Polarization curves of PW-Co₃N NWA/NF before and after CV testing of 1000 and 5000 cycles. The inset is corresponding Nyquist plots. (d) The chronoamperometric test recorded at overpotential of 92 mV.

Supplementary Figure 19. C_{dl} values and cyclic voltammogram curves. (a) C_{dl} values of different materials; CV curves of (b) PW-Co₃N NWA/NF, (c) W-Co₃N NWA/NF, (d) P-Co₃N NWA/NF, (e) Co₃N NWA/NF, (f) Ni foam, (g) PW-Co-precursor and (h) Pt/C in the double layer capacitive region at the scan rates from 10 mV to 100 mV s^{-1} .

Supplementary Figure 20. Nyquist plots of PW-Co₃N NWA/NF, W-Co₃N NWA/NF, P-Co₃N NWA/NF, Co₃N NWA/NF, Ni foam, PW-Co-precursor and Pt/C obtained at overpotential of 100 mV for HER. The inset is the enlarged view of Nyquist plots of PW-Co₃N NWA/NF, W-Co₃N NWA/NF, P-Co₃N NWA/NF, Co₃N NWA/NF, Ni foam, PW-Co-precursor and Pt/C.

Supplementary Table 2. Comparison of the electrocatalytic activities of PW-Co₃N NWA/NF with other reported materials for HzOR.

Materials	electrolyte	J (mA cm ⁻²)	Potential (mV)	Reference
		10	-55	
PW-Co ₃ N NWA/NF	1.0 M KOH+0.1 M N ₂ H ₄	50	-29	This work
		200	27	
Co ₃ Ta/C	3.0 M KOH+0.5 M N ₂ H ₄	25.2	60	Nat. Commun. 10 , 4514 (2019).
Cu ₁ Ni ₂ -N	1.0 M KOH+0.5 M N ₂ H ₄	10	0.5	Adv. Energy Mater. 9 , 1900390 (2019).
Ni _x P/NF	1.0 M NaOH+0.1 M N ₂ H ₄	172	100	Appl. Catal. B: Environ. 241 , 292-298 (2019).

Rh/N-CBs	1.0 M KOH+0.05 M N ₂ H ₄	10	72	ACS Appl. Mater. Interfaces 11 ,35039-3504 (2019).
Fe-CoS₂	1.0 M KOH+0.1 M N ₂ H ₄	100	129	Nat. Commun. 9 , 4365 (2018).
CoSe₂	1.0 M KOH+0.5 M N ₂ H ₄	10	-17	Angew. Chem. Int. Ed. 57 , 7649-7653 (2018).
Ni₃S₂/NF	1.0 M KOH+0.2 M N ₂ H ₄	100	415	J. Mater. Chem. A 6 , 19201-19209 (2018)
Ni₂P/NF	1.0 M KOH+0.5 M N ₂ H ₄	50	-25	Angew. Chem. Int. Ed. 56 , 842-846 (2017).
Ni-NSA	3.0 M KOH+1.0 M N ₂ H ₄	227.6	250	Angew. Chem. Int. Ed. 55 , 693-697 (2016).
NiZn	1.0 M KOH+1.0 M N ₂ H ₄	320	600	Angew. Chem. Int. Ed. 53 , 10336-0339 (2014).

Supplementary Table 3. Comparison of the electrocatalytic activities of PW-Co₃N NWA/NF with recently reported transition metal nitride for HER in 1.0 M KOH.

Materials	η_{10} (mV)	Tafel slope (mV dec⁻¹)	Reference
PW-Co₃N NWA/NF	41	40	This work
NiMoN/NF	56	45.6	Nat. Commun. 10 , 5106 (2019).
NiCoN/C nanocages	103	-	Adv. Mater. 31 , 1805541 (2019).

Ni ₃ N/C	64	48	Angew. Chem. Int. Ed. 58 , 1-6 (2019).
Co ₂ N/Co/CF	12	41.6	ACS Energy Lett. 4 , 1594–1601 (2019).
V-Co ₄ N/NF	37	44	Angew. Chem. Int. Ed. 57 , 5076-5080 (2018).
Co-Ni ₃ N	194	156	Adv. Mater. 30 , 1705516 (2018).

Supplementary Table 4. Comparison of OHZS performance of PW-Co₃N NWA/NF with other work.

Materials	electrolyte	J (mA cm ⁻²)	cell voltage (mV)	Reference
PW-Co₃N NWA/NF	1.0 M KOH+0.1 M N ₂ H ₄	10	25	This work
		200	277	
Cu₁Ni₂-N	1.0 M KOH+0.5 M N ₂ H ₄	10	240	Adv. Energy Mater. 2019, 9 , 1900390
Fe-CoS₂	1.0 M KOH+0.1 M N ₂ H ₄	100	610	Nat. Commun. 2018, 9 , 4365
CoSe₂	1.0 M KOH+0.5 M N ₂ H ₄	10	164	Angew. Chem. Int. Ed. 2018, 57 , 7649 -7653
Ni(Cu)	1.0 M KOH+0.5 M N ₂ H ₄	200	641	ACS Sustainable Chem. Eng. 2018, 6 , 12746-12754

Ni₂P/NF	1.0 M KOH+0.5 M N ₂ H ₄	100	450	Angew. Chem. Int. Ed. 2017, 56 ,842-846
---	-----	-----	--

Supplementary Table 5. Comparison of DHzFC performance of PW-Co₃N NWA/NF with other work.

Materials	Anodic fuel	temperatures	P_{max} (mW cm⁻²)	OCV (V)	Reference
PW-Co₃N NWA/NF	1.0 M KOH+0.5 M N ₂ H ₄	room temperature	46.3	0.98	This work
nanostructured Cu film	4.0 M NaOH+20 wt.% N ₂ H ₄	room temperature	29.1	about 1	Adv. Mater. 2015, 27 , 2361-2366
NPGLs	4.0 M NaOH+10 wt.% N ₂ H ₄	40 °C	42.5	0.89	Sci. Rep. 2012, 2 ,941
Pt₅₃Cu₄₇/C	1.0 M NaOH+1.0 M N ₂ H ₄	60 °C	32.6	about 0.8	Appl. Catal. B: Environ. 2018, 236 ,36-44
Co-PPy/C	1.0 M KOH+5 wt.% N ₂ H ₄	50 °C	75	0.73	J. Electrochem. Soc. 2014, 161 , F889-893

5. The possible contribution of Ni foam should be noticed. The authors should be aware that nickel compounds such as Ni₂P show very good electrocatalytic activity in this system.

Our response: We thank the reviewer's professional question. It is true that nickel compounds like Ni₂P has been reported possessing excellent catalytic activity for both HzOR and HER (*Angew. Chem. Int. Ed.* 2017, **56**, 842-846). However, in our PW-Co₃N NWA/NF sample, the XRD results (Supplementary Fig. 2) indicated that there is no signal for CoP or Ni₂P phase. Moreover, the atomic ratio in PW-Co₃N/NF for P: W: Co is confirmed to be 1: 3.50: 36.9 by

EDS and ICP-AES measurements, which means that it is reasonable to neglect the contribution of metal phosphide even assuming the existence of possible trace amount of metal phosphides. In addition, in order to further demonstrate the intrinsic catalytic activity toward HzOR and HER of PW-Co₃N NWA/NF, we performed extra control experiments using sodium hypophosphite with Ni foam annealed at 420 °C for 2 h under NH₃ atmosphere. The existence of phosphorus and nitride in the prepared material (denoted as PN/NF) was confirmed via EDS spectrum (added as Supplementary Fig. 21). The as-obtained PN-Ni foam exhibits much poorer HzOR and HER performance compared with that of PW-Co₃N NWA/NF (Added as Supplementary Fig. 22), further proving that the possible effects of nickel compound can be excluded. The related discussion has been added in Lines 10-15, Page 17.

Supplementary Figure 2. XRD pattern of PW-Co₃N NWA/NF.

Supplementary Figure 21. EDS spectrum of PN/NF.

Supplementary Figure 22. HzOR and HER performance of PN/NF and PW-Co₃N NWA/NF. HzOR: (a) LSV curves, (b) Nyquist plots; HER: (c) LSV curves, (d) Nyquist plots.

6. A recent benchmark (Nature Commun. 2019, 10, 1-9, Article number: 4514) related to hydrazine electrooxidation should be cited.

Our response: We thank the reviewer for the good suggestion. As you suggested, we have added this literature as Ref. 24 of the reference part and emphasized its importance in the revised manuscript. Please Lines 7-9, Page 5 of the revised manuscript.

7. Many typos exist, such as "the mixer of H₂ and O₂", "a ultrasmall", ...

Our response: We thank the reviewer for pointing out these errors for us. We have carefully re-checked our manuscript and corrected the grammatical errors and typos, which are highlighted in the main text of the revised manuscript.

Reviewer #3 (Remarks to the Author):

The authors proposed a facile two-step strategy to synthesize P, W-codoped cobalt nitride nanowire array material. They characterized the structure and morphology by a number of techniques such as TEM, SEM, XPS, and XRD. They further demonstrated the potential application as bifunctional electrocatalyst for hydrazine oxidation and hydrogen evolution. The catalytic mechanism has been studied by density function theory calculation. This work is publishable in Nature Communications after addressing the following concerns:

Our response: We thank the reviewer for the encouraging comments on our manuscript. It is greatly appreciated for the reviewer to provide constructive suggestions, which is quite helpful and significant for us to further improve the quality of the manuscript. In the revised manuscript, we have conducted a series of additional experiments and provided related discussions following your comments and suggestions.

(1) In Figure 5a, the comparison is not fair because of different thermodynamic potentials of OWS (1.23 V) and OH₂S (-0.33 V). I suggest the comparison of faraday efficiencies and overpotentials.

Our response: We thank the reviewer for the professional suggestion. As you suggested, we calculated the corresponding overpotentials in Fig. 6b of the revised manuscript considering the thermodynamic cell voltage of 1.23 V for OWS and -0.33 V for OH₂S. Specifically, it only requires the overpotentials (compared to the theoretical value of -330 mV) of 358, 428, 501 and 607 mV in OH₂S system to reach current densities of 10, 50, 100 and 200 mA cm⁻² (Fig. 6b), respectively, while much higher (compared to the theoretical value of 1230 mV) overpotential of 350, 530, 650 and 869 mV are required in the case of OWS to obtain the same current density. The results proved that our OH₂S system not only demands less electric energy but also exhibits the feasible kinetics considering the thermodynamic potentials for H₂ production. The Faraday efficiency of OWS is measured to be about 94% (Fig. 6d), which is slightly lower than that of

OHzS system (96%). The related discussions have also been added to the main text of the revised manuscript. Please see Line 22, Page 17 and Lines 9-15, Page 18 in the revised manuscript.

Figure 6. Electrochemical performance of OHzS using PW-Co₃N NWA/NF without DHzFC: (a) Comparison of LSV curves for OHzS in 1.0 M KOH/0.1 M hydrazine and OWS in 1.0 M KOH using PW-Co₃N NWA/NF as both the anode and cathode, (b) Comparison of the overpotentials required to reach different current densities for OHzS and OWS, (c) The amount of hydrogen theoretically calculated and experimental measured for OHzS, (d) The amount of hydrogen theoretically calculated and experimental measured for OWS, (e) I-t curve of OHzS recorded at 98 mV for 20 h.

(2) A table should be given to compare the activity of HzOR and HER with the state-of-the-art literature.

Response: We thank the reviewer for the good suggestion. According to your suggestion, we added two tables (Supplementary Table 2 and Table 3 of the revised manuscript) to compare our results with recently reported state-of-the-art literatures for HzOR and HER activities, respectively, which are also provided below for your convenience.

Supplementary Table 2. Comparison of the electrocatalytic activities of PW-C₀₃N NWA/NF with other reported materials for HzOR.

Materials	electrolyte	J (mA cm ⁻²)	Potential (mV)	Reference
		10	-55	
PW-C₀₃N NWA/NF	1.0 M KOH+0.1 M N ₂ H ₄	50	-29	This work
		200	27	
C₀₃Ta/C	3.0 M KOH+0.5 M N ₂ H ₄	25.2	60	Nat. Commun. 10 , 4514 (2019).
Cu₁Ni₂-N	1.0 M KOH+0.5 M N ₂ H ₄	10	0.5	Adv. Energy Mater. 9 , 1900390 (2019).
Ni_xP/NF	1.0 M NaOH+0.1 M N ₂ H ₄	172	100	Appl. Catal. B: Environ. 241 , 292-298 (2019).
Rh/N-CBs	1.0 M KOH+0.05 M N ₂ H ₄	10	72	ACS Appl. Mater. Interfaces 11 ,35039-3504 (2019).
Fe-CoS₂	1.0 M KOH+0.1 M N ₂ H ₄	100	129	Nat. Commun. 9 , 4365 (2018).
CoSe₂	1.0 M KOH+0.5 M N ₂ H ₄	10	-17	Angew. Chem. Int. Ed. 57 , 7649-7653 (2018).
Ni₃S₂/NF	1.0 M KOH+0.2 M N ₂ H ₄	100	415	J. Mater. Chem. A 6 , 19201- 19209 (2018)
Ni₂P/NF	1.0 M KOH+0.5 M N ₂ H ₄	50	-25	Angew. Chem. Int. Ed. 56 , 842-846 (2017).

Ni-NSA	3.0 M KOH+1.0 M N ₂ H ₄	227.6	250	Angew. Chem. Int. Ed. 55 , 693-697 (2016).
NiZn	1.0 M KOH+1.0 M N ₂ H ₄	320	600	Angew. Chem. Int. Ed. 53 , 10336-0339 (2014).

Supplementary Table 3. Comparison of the electrocatalytic activities of PW-CO₃N NWA/NF with recently reported transition metal nitride for HER in 1.0 M KOH.

Materials	η_{10} (mV)	Tafel slope (mV dec⁻¹)	Reference
PW-CO₃N NWA/NF	41	40	This work
NiMoN/NF	56	45.6	Nat. Commun. 10 , 5106 (2019).
NiCoN/C nanocages	103	-	Adv. Mater. 31 , 1805541 (2019).
Ni₃N/C	64	48	Angew. Chem. Int. Ed. 58 , 1-6 (2019).
Co₂N/Co/CF	12	41.6	ACS Energy Lett. 4 , 1594–1601 (2019).
V-CO₄N/NF	37	44	Angew. Chem. Int. Ed. 57 , 5076- 5080 (2018).
Co-Ni₃N	194	156	Adv. Mater. 30 , 1705516 (2018).

(3) What is the efficiency from hydrazine to hydrogen for self-powered H₂ production system?

The authors should compare it with other hydrogen generation systems.

Our response: We thank the reviewer for the constructive suggestion. Following your suggestion, we have calculated the efficiency from hydrazine to hydrogen in our self-power H₂ production system. The total efficiency is calculated according to the following equation:

$$\text{Total Efficiency (TE, \%)} = N_{\text{H}_2} / 2N_{\text{DHZFC}} * \text{FE}_{\text{OHZS}} * 100 \%$$

Where N_{H₂} is the amount (mol) of produced hydrogen, N_{DHzFC} is the amount (mol) of consumed hydrazine from the electrolyte of DHzFC and FE is the Faraday efficiency (%) of OHzS. The amount (mol) of the consumed hydrazine in the DHzFC were carefully measured by a UV-vis spectrophotometric method proposed by Watt and Chrisp (*Adv. Mater.* 2019, **31**, 1902709; *ACS Catal.* 2019, **9**, 7311-7317; *Anal. Chem.* 1952, **24**, 2006-2008). The color reagent is the mixed solution of 1.0 g p-(dimethylamino) benzaldehyde, 50 mL ethanol and 5 mL of 0.12 M HCl. Meanwhile, firstly, 50 μL of electrolyte from DHzFC after reaction was added into measuring flask and DIW was also added into the same measuring flask till 500 mL in total volume. Then 1 mL of the above solution from the measuring flask was mixed with 1 mL color reagent and 4 mL of 0.12 M HCl. After standing at room temperature for 20 min, the UV-vis spectrum of the solution was collected. The concentration-absorbance curves were calibrated using standard hydrazine solution in a series of concentrations (as indicated in Supplementary Fig. 24a-b). Following this strategy, the total efficiency is calculated to be about 45.8 % in our system, which is outstanding compared to other hydrogen generation system, such as mechanical energy driven self-power system (43.8%, *Adv. Mater.* 2015, **27**, 272-276), hybrid energy cell (16%, *Energy Environ. Sci.*, 2013, **6**, 2429-2434) and water photolysis (12.3%, *Science*, 2014, **345**, 1593). The UV-Vis absorption spectra of different concentration of standard hydrazine, corresponding calibration curve and the UV-Vis absorption spectra of diluted electrolyte taken from DHzFC were provide below for your convenience. The related results are also provided as

Supplementary Fig. 24 in the supplementary information of the revised manuscript. The related discussion has been added into main text of the revised manuscript, please see Lines 18-20, Page 19.

Supplementary Figure 24. Detection of hydrazine using UV-vis spectrophotometric method:

(a) UV-vis absorption spectra of different concentrations of hydrazine stained with color reagent, (b) calibration curve for calculating the hydrazine concentration, (c) UV-vis absorption spectra of the diluted electrolytes from DHzFC stained with color reagent.

(4) The authors described that their catalyst has a “Pt-like” activity. They may need to compare the activity of HER with commercial Pt/C. Further, DFT calculation of HER free energy for Pt should also be given for comparison.

Our response: We thank the reviewer for the good suggestion. According to the suggestion, we added the HER performance of commercial 20 wt.% Pt/C loaded on Ni foam with the same mass loading of 2 mg cm^{-2} (Fig. 5a, b in the revised manuscript). As indicated below, the overpotential of PW- Co_3N NWA/NF is only 9 mV larger than that of Pt/C (32 mV) at 10 mA cm^{-2} , and the Tafel slope of PW- Co_3N NWA/NF is comparable to benchmark Pt/C (31 mV dec^{-1}). The related discussions were added in the main text of the revised manuscript (Please see Lines 16 and 22-23, Page 15; Line 13, Page 16). Moreover, according to your suggestion, we calculated ΔG_{H^*} value is -0.10 eV on Pt (111) as shown in Fig. 8e, which is consistent with previous studies (*J. Electrochem. Soc.* 2005, **152**(2), J23-J26). Please see the related discussions at Lines 11-12, Page 22 of the revised manuscript.

Figure 5. Electrocatalytic activity towards HER in 1.0 M KOH electrolyte. (a) Polarization curves of PW-Co₃N NWA/NF, W-Co₃N NWA/NF, P-Co₃N NWA/NF, Co₃N NWA/NF, bare Ni foam, PW-Co-precursor, and Pt/C towards HER, (b) The corresponding Tafel plots derived from (a), (c) Polarization curves of PW-Co₃N NWA/NF before and after CV testing of 1000 and 5000 cycles. The inset is corresponding Nyquist plots. (d) The chronoamperometric test recorded at overpotential of 92 mV.

Figure 8. DFT calculated profiles of free energy: Top- and side- view of atomic structure models for (a, c) Co₃N; (b, d) PW-Co₃N; (e) The free energy diagram of HER at the equilibrium potential for Co₃N and PW-Co₃N, H* denotes that intermediate adsorbed hydrogen; (f) The d band of density of states (DOS) of Co₃N and PW-Co₃N; (g) Water adsorption energy on Co₃N and PW-Co₃N; Top-(h) and side-(i) view of charge density difference analysis for PW-Co₃N with the cyan region representing charge depletion and the yellow region representing charge accumulation. The isosurface value is 0.012 eÅ⁻³; (j) The free energy profiles of HzOR on the Co₃N and PW-Co₃N surfaces. The inset in (j) are the most stable configurations of the each adsorbed intermediate on the Co site.

(5) Hydrazine is a highly toxic chemical. Therefore, the authors should discuss further the potential problems and challenges for the large-scale applications of as-proposed self-powered H₂ production system.

Our response: We thank the reviewer for the valuable suggestion. According to your suggestion, we have added the related discussions about the potential challenges and solutions on the high toxicity nature of the involved hydrazine fuel. Please see Lines 20-22, Page 19 and Lines 9-14, Page 20 of the revised manuscript.

We also provide the content below for your convenience:

Compared with gaseous hydrogen or carbon monoxide, hydrazine has the advantage of more convenient transportation and storage as a liquid fuel at ordinary temperatures. However, it is necessary to state that hydrazine is a highly toxic chemical, which may be a challenging issue for the large-scale applications (*Energy Environ. Sci.* 2011, **4**, 1255–1260). This is also the major reason that we choose to build our self-powered system working at room temperature and low hydrazine concentration. In order to tackle this obstacle, Asazawa et al. (*Angew. Chem. Int. Ed.* 2007, **46**, 8024-8027) designed a detoxification technique to fix the hydrazine as the carbonyl groups in the harmless, stable and recyclable polymer for storage, which can be released using water or KOH (aq) when needed. This strategy has been considered as one of the promising strategies for large-scale applications involving toxic hydrazine (*Angew. Chem. Int. Ed.* 2014, **53**, 10336 -10339; *Nat. Commun.* 2018, **9**, 4365).

Reviewers' comments:

Reviewer #1 (Remarks to the Author):

The authors have addressed most of the previous comments, except for 5. On page 16, it mentioned "energetically favorable to replace surface Co atoms ...". Please provide those energetic data.

The text now appears on page 21.

"it is energetically favorable to replace surface Co atoms in Co₃N with W atoms and subsurface N atoms with P atoms"

Please provide the substitution energy of W to Co and the substitution energy of N by P (by comparing the total energy of difference systems and adopting proper reference for W, Co, N and P).

I would recommend publication of the manuscript if the requested information is added.

Reviewer #2 (Remarks to the Author):

In general, the authors have substantially improved their manuscript by conducting in-depth studies of their electrocatalysts, and the reviewer recommends the acceptance of this work to our journal. However, I still believe it is not necessary to overclaim their results by selectively narrowing the comparison to their own advantages. For instance, in the Supplementary Table 5, the authors might have intentionally neglected those professional discussions for high performance DHZFCs. For example, in the work by Lao et al. at J. Power Sources 2010, 195, 4135, the maximum power reaches ~1000 mW cm⁻² with an OCV of 1.75 V, which is much much better than the observed 46.3 mW cm⁻² and 0.98 V in the present work. The authors might argue their data were collected under different conditions such as at room temperature. This is because most fuel cells are designed to work only under specific conditions, such as 80 C. Besides, the authors weakened the fact that their PW-Co₃N NWA/NF sample has the highest capacitance (which is directly relevant to their surface areas), more than twice of that of Co₃N NWA/NF, and nearly two orders of magnitude larger than Ni foam and PW-Co-precursor. How to provide a fair comparison may need protocols or standards, which are yet to be established for this community. I remind the editors to notice a recent discussion by Martin Pumera at ACS Nano entitled "Will Any Crap We Put into Graphene Increase Its Electrocatalytic Effect?", which unfortunately criticized the massive irresponsible researches regarding doping and co-doping.

Reviewer #3 (Remarks to the Author):

The authors have addressed my concerns by providing additional results and discussions. This paper is acceptable for publication in Nat Common

Response to Reviewers' Comments

Reviewer #1 (Remarks to the Author):

The authors have addressed most of the previous comments, except for 5. On page 16, it mentioned “energetically favorable to replace surface Co atoms ...”. Please provide those energetic data. The text now appears on page 21. "it is energetically favorable to replace surface Co atoms in Co_3N with W atoms and subsurface N atoms with P atoms". Please provide the substitution energy of W to Co and the substitution energy of N by P (by comparing the total energy of difference systems and adopting proper reference for W, Co, N and P). I would recommend publication of the manuscript if the requested information is added.

Our response: We really appreciate the reviewer’s valuable suggestions and support on publication of our work! According to your suggestion, we provided the energetic data of the two different models in the revised Supplementary Fig. 26, which is also provided below for your convenience. In our model, there are three Co atom layers in Co_3N (001) surface. We calculated 4W atoms substituted for 4 Co atoms in different Co layers of Co_3N (001) surface. After geometric optimization, as shown in Supplementary Fig. 26a, total energy of 4 W substituted for 4 Co atom at first Co layers of Co_3N (001) surface are -330.180 eV, and its structure shows no significant distortion. Contrastively, there will be serious structural distortions for the system with 4 W doped at second and third Co layers of Co_3N (001) planes where the N atoms of second layer totally deviate from its original position, as shown in Supplementary Fig. 26b-c. This is not reasonable since the crystal structure can be well maintained after W doping according to the XRD results, although the total energy (-335.152 eV for doping second Co layer, -332.196 eV for doping third Co layer) are lower compared to the system with surface doping model. Therefore, 4 W substituted for 4 Co atom at first Co layers of Co_3N (001) surface (denoted as sur-4W: Co_3N) is the most favorable configuration than that of the other Co layers.

Secondly, we would like to check the substitution doping of a single P atom in different N layer of sur-4 W: Co_3N surface. After geometric optimization, as shown in Fig. 26d-f, the total energy of P substituted for N atom at first, second and third N layers of sur-4 W: Co_3N surface are calculated to be -328.121 eV, -327.828 eV, and -328.103 eV, respectively. Their structures show no significant distortion which are consistent with the XRD results after P/W doping. Thus the results indicate that the P is favorable to replace N atom at first N layers of sur-4 W: Co_3N surface due to the lowest energy configuration. Based on these results, we adopt the optimal structure of replace surface Co atoms in Co_3N (001) with W atoms and subsurface N atom with P atom model during our calculation.

Supplementary Figure 26. Model simulation of 4 W atoms and 1 P atom substituted Co_3N . Model simulation of 4 W atoms substituted Co atoms at (a) first, (b) second and (c) third Co layers of Co_3N (001) surface, after geometric optimization; Model simulation of 1 P atom substituted N atom at (d) first, (e) second and (f) third N layers of sur-4 W: Co_3N surface, after geometric optimization. The balls in yellow, pink, green and purple represent Co, W, P and N atoms, respectively.

Reviewer #2 (Remarks to the Author) :

In general, the authors have substantially improved their manuscript by conducting in-depth studies of their electrocatalysts, and the reviewer recommends the acceptance of this work to our journal. However, I still believe it is not necessary to overclaim their results by selectively narrowing the comparison to their own advantages. For instance, in the Supplementary Table 5, the authors might have intentionally neglected those professional discussions for high performance DHzFCs. For example, in the work by Lao et al. at J. Power Sources 2010, 195, 4135, the maximum power reaches $\sim 1000 \text{ mW cm}^{-2}$ with an OCV of 1.75 V, which is much much better than the observed 46.3 mW cm^{-2} and 0.98 V in the present work. The authors might argue their data were collected under different conditions such as at room temperature. This is because most fuel cells are designed to work only under specific conditions, such as $80 \text{ }^\circ\text{C}$. Besides, the authors weakened the fact that their PW-Co₃N NWA/NF sample has the highest capacitance (which is directly relevant to their surface areas), more than twice of that of Co₃N NWA/NF, and nearly two orders of magnitude larger than Ni foam and PW-Co-precursor. How to provide a fair comparison may need protocols or standards, which are yet to be established for this community. I remind the editors to notice a recent discussion by Martin Pumera at ACS Nano entitled "Will Any Crap We Put into Graphene Increase Its Electrocatalytic Effect?", which unfortunately criticized the massive irresponsible researches regarding doping and co-doping.

Our response: We really thank the reviewer's valuable comments and support on the publication of our work! According to suggestion, we change the words/phrases such as "best", "record high" to "decent", "competitive" or "remarkable" in both Introduction (Page 6) and Result (Page 19) parts in order to avoid unnecessary overclaim, which are highlighted in the revised manuscript.

We totally agree with the reviewer's opinion that "how to provide a fair comparison may need protocols or standards, which are yet to be established for this community", since it will be quite difficult to illustrate the experimental data to readers. For instance, as the work mentioned above by the reviewer (*J. Power Sources* 2010, **195**, 4135), the maximum power density can achieve as high as 1 W cm⁻² under their conditions by integrating 20 wt.% H₂O₂ in 5 wt.% H₂SO₄ solution as oxidizer and 10 wt.% N₂H₄ in 15 wt.% NaOH solution at 80 °C. In our work, we adopt the room temperature as working environment is based on the concept that the fewer demand of external equipment could make the extend application of "self-power" system be more satisfied to the need of actual large-scale application thus avoid the heating equipment (it is only the proof-of-concept for sure). Besides, the lower working temperature could reduce the risk of highly toxic hydrazine.

As for the comparison standards, we agree that fair comparison standards for this community need the joint efforts of all the researches in this field including us obviously, and in our present work, the comparison standards are referring to several recent reported literatures, where the prepared materials on Ni foam is used as electrode directly and the current density is calculated via the geometric area of materials (*J. Am. Chem. Soc.* 2019, **141**, 7537–7543; *Nat. Commun.* 2018, **9**, 924; *Nat. Commun.* 2018, **9**, 4531; *Angew. Chem. Int. Ed.* 2018, **57**, 7649-7653; *Angew. Chem. Int. Ed.* 2017, **56**, 842-846;). As a conclusion, it is worth for us to reiterate that the most important scientific contribution of our work is the first report demonstrating the notable effect of heteroatom doping of transition metal nitride can largely improve its HzOR activity, and simultaneously enhance the HER activity in certain degree. Importantly, the DFT calculation combined with systematic experimental data unravels the fundamental origin of the enhanced performance. We are confident that our work regarding the P/W doped Co₃N nanowire array electrode possesses sufficient novelty regarding the reported strategy for simultaneous enhancement for HzOR and HER activity, as well as significant scientific contribution regarding

the fundamental understanding over the structure-property relationship based on the combined DFT calculation and experimental data.

Reviewer #3 (Remarks to the Author):

The authors have addressed my concerns by providing additional results and discussions. This paper is acceptable for publication in Nat Common.

Our response: We greatly appreciate the reviewer for the support on the publication of our work!

REVIEWERS' COMMENTS:

Reviewer #1 (Remarks to the Author):

The author provided additional structural figures and total energy for the structures. They did not address my previous questions for calculating substitution energy. The total energy does not support their claim on page 21 that "it is energetically favorable to replace surface Co atoms in Co₃N with W atoms and subsurface N atoms with P atoms".

They would need to perform calculations to compare energy difference to find out about whether it is "energetically favorable to replace". For example, using the following equation:

$$E(\text{W-doped Co}_3\text{N}) - E(\text{Co}_3\text{N}) + E(\text{Co metal}) - E(\text{W metal}).$$

If they just want to express where the substitution would be in the slab, they can rewrite that sentence. Even there, please be cautious about using "energetically favorable", because the W on the surface is higher in energy than in the subsurface ones.

Response to Reviewers' Comments

Reviewer #1 (Remarks to the Author):

The author provided additional structural figures and total energy for the structures. They did not address my previous questions for calculating substitution energy. The total energy does not support their claim on page 21 that "it is energetically favorable to replace surface Co atoms in Co_3N with W atoms and subsurface N atoms with P atoms". They would need to perform calculations to compare energy difference to find out about whether it is "energetically favorable to replace". For example, using the following equation: $E(\text{W-doped Co}_3\text{N}) - E(\text{Co}_3\text{N}) + E(\text{Co metal}) - E(\text{W metal})$. If they just want to express where the substitution would be in the slab, they can rewrite that sentence. Even there, please be cautious about using "energetically favorable", because the W on the surface is higher in energy than in the subsurface ones.

Our response: We really thank the reviewer's valuable suggestions. It is true that our statement of "energetically favorable" may be inappropriate under this circumstance. According to your suggestion, we have re-written the sentence as "Combined with the theoretical calculation and experimental result regarding the structural stability after doping, it is reasonable to use the model where the surface Co atoms in Co_3N are replaced by W atoms and subsurface N atoms are replaced by P atoms, as indicated in Fig. 8a-d, Supplementary Fig. 26 and Supplementary Note 2.", which is also highlighted in the revised manuscript (Page 21).